# Abundant toxin-related genes in the genomes of beneficial symbionts from deep-sea hydrothermal vent mussels

Lizbeth Sayavedra[1], Manuel Kleiner[1†], Ruby Ponnudurai[2†], Silke Wetzel[1], Eric Pelletier[3,4,11], Valerie Barbe[3], Nori Satoh[5], Eiichi Shoguchi[5], Dennis Fink[1], Corinna Breusing[6], Thorsten BH Reusch[6], Philip Rosenstiel[7], Markus B Schilhabel[7], Dörte Becher[8,9], Thomas Schweder[2,8], Stephanie Markert[2,8], Nicole Dubilier[1,10], Jillian M Petersen[1*‡]

[1]Max Planck Institute for Marine Microbiology, Bremen, Germany; [2]Institute of Pharmacy, Ernst-Moritz-Arndt-University, Greifswald, Germany; [3]Genoscope - Centre National de Séquençage, Commissariat à l'énergie atomique et aux énergies alternatives, Evry, France; [4]Metabolic Genomics Group, Commissariat à l'énergie atomique et aux énergies alternatives, Evry, France; [5]Marine Genomics Unit, Okinawa Institute of Science and Technology, Onna, Japan; [6]Evolutionary Ecology, GEOMAR Helmholtz Centre for Ocean Research Kiel, Kiel, Germany; [7]Institute of Clinical Molecular Biology, Kiel, Germany; [8]Institute of Marine Biotechnology, Greifswald, Germany; [9]Institute of Microbiology, Ernst-Moritz-Arndt-University, Greifswald, Germany; [10]University of Bremen, Bremen, Germany; [11]University of Évry-Val d'Essonne, Evry, France

*For correspondence:
jmpeters@mpi-bremen.de

†These authors contributed equally to this work

Present address: ‡Department of Microbiology and Ecosystem Science, Division of Microbial Ecology, Research Network Chemistry Meets Microbiology, University of Vienna, Vienna, Austria

Competing interests: The authors declare that no competing interests exist.

**Abstract** *Bathymodiolus* mussels live in symbiosis with intracellular sulfur-oxidizing (SOX) bacteria that provide them with nutrition. We sequenced the SOX symbiont genomes from two *Bathymodiolus* species. Comparison of these symbiont genomes with those of their closest relatives revealed that the symbionts have undergone genome rearrangements, and up to 35% of their genes may have been acquired by horizontal gene transfer. Many of the genes specific to the symbionts were homologs of virulence genes. We discovered an abundant and diverse array of genes similar to insecticidal toxins of nematode and aphid symbionts, and toxins of pathogens such as *Yersinia* and *Vibrio*. Transcriptomics and proteomics revealed that the SOX symbionts express the toxin-related genes (TRGs) in their hosts. We hypothesize that the symbionts use these TRGs in beneficial interactions with their host, including protection against parasites. This would explain why a mutualistic symbiont would contain such a remarkable 'arsenal' of TRGs.

## Introduction

Mussels of the genus *Bathymodiolus* dominate deep-sea hydrothermal vents and cold seeps worldwide. The key to their ecological and evolutionary success is their symbiosis with chemosynthetic bacteria that provide them with nutrition (*von Cosel, 2002*; *Van Dover et al., 2002*). *Bathymodiolus* mussels host their symbionts inside specialized gill epithelial cells called bacteriocytes (*Cavanaugh et al., 2006*; *Petersen and Dubilier, 2009*).

Their filtering activity exposes *Bathymodiolus* mussels to a plethora of diverse microbes in their environment. Despite this, they are colonized by only one or a few specific types of chemosynthetic symbionts. Some mussel species associate exclusively with sulfur-oxidizing (SOX) symbionts that use

**eLife digest** Although bacteria are commonly associated with causing illness, many are actually beneficial to the organism they live in or on. The phenomenon of one species helping another to survive is known as symbiosis.

Animals thrive at hydrothermal vents in the deep sea because of their partnerships with symbiotic bacteria. The bacteria use the geochemical energy found at hydrothermal vents to convert carbon into sugars, thus providing their animal hosts with essential nutrients. Unlike the symbiotic communities that associate with humans and other mammals, in which thousands of bacterial species co-exist, deep-sea mussels associate with just one or two species of symbiotic bacteria. This relative simplicity is ideal for investigating how the intimate associations between animals and bacteria work.

Genes contain the instructions cells and organisms need to survive, and so one way that researchers investigate symbiosis is by studying the genes of the organisms involved. Such studies of beneficial bacteria are beginning to reveal that the molecular mechanisms involved in symbiosis are remarkably similar to those responsible for the harmful effects produced by some bacteria.

By performing genetic sequencing on the symbiotic bacteria from deep-sea mussels, Sayavedra et al. have discovered that the bacteria have an unusually large number of toxin-like genes, and that all of these genes are active in the bacteria when they are inside host mussels. This was unexpected, as the bacteria are known to benefit their mussel hosts. The toxin-like genes from the symbiotic bacteria are similar to toxins found in the bacteria that cause diseases such as cholera and the plague in humans and other animals.

Sayavedra et al. suggest that the symbiotic bacteria have 'tamed' these toxins to use them in beneficial interactions with their host. For example, some of the toxins could help the bacteria and mussels to recognize and interact with each other, and others could help to protect the mussel host from its natural enemies. The next step will be to test these ideas, which will be challenging as the mussels cannot be bred in the laboratory.

reduced sulfur compounds and sometimes hydrogen as an energy source, and carbon dioxide as a carbon source. Some have only methane-oxidizing (MOX) symbionts that use methane as an energy source and carbon source. Some mussel species host both types in a dual symbiosis (*Fisher et al., 1993*; *Distel et al., 1995*; *Duperron et al., 2006*; *Dubilier et al., 2008*; *Petersen et al., 2011*). In all species except one, a single 16S rRNA phylotype for each type of symbiont (SOX or MOX) is found in the gills (*Dubilier et al., 2008*). There are more than 30 described *Bathymodiolus* species, and most associate with a characteristic symbiont phylotype, which is not found in other species (*Duperron et al., 2013*).

Although these associations are clearly very specific, the molecular mechanisms that underpin this specificity are still unknown. No chemosynthetic symbiont has ever been obtained in pure culture. Therefore, molecular methods for investigating uncultured microbes have been essential for understanding their biodiversity, function, and evolution (reviewed by *Dubilier et al., 2008*).

The *Bathymodiolus* symbionts are assumed to be horizontally transmitted, which means that each new host generation must take up their symbionts from the surrounding environment or co-occurring adults (*Won et al., 2003b*; *Kadar et al., 2005*; *DeChaine et al., 2006*; *Fontanez and Cavanaugh, 2014*; *Wentrup et al., 2014*). To initiate the symbiosis, hosts and symbionts must have evolved highly specific recognition and attachment mechanisms. Once they have been recognized, the symbionts need to enter host cells and avoid immediate digestion, just like other intracellular symbionts such as *Burkholderia rhizoxinica* and *Rhizobium leguminosarum*, or pathogens such as *Legionella*, *Listeria*, or *Yersinia* (*Hentschel et al., 2000*; *Moebius et al., 2014*). Indeed, like many intracellular pathogens, the *Bathymodiolus* symbionts seem to induce a loss of microvilli on the cells they colonize (*Cossart and Sansonetti, 2004*; *Bhavsar et al., 2007*; *Haglund and Welch, 2011*; *Wentrup et al., 2014*). Finally, the symbionts achieve dense populations inside the host cells (e.g., *Duperron et al., 2006*; *Halary et al., 2008*). Therefore, they must be able to avoid immediate digestion by their hosts. Although the mechanisms of host cell entry and immune evasion have been extensively studied in pathogens and plant–microbe associations such as the rhizobia-legume symbiosis, far less is known about the mechanisms beneficial symbionts use to enter and survive within animal host cells.

The symbiosis between *Vibrio fisheri* bacteria and *Euprymna scolopes* squid is one of the few beneficial host-microbe associations where the molecular mechanisms of host-symbiont interaction have been investigated. A number of factors are involved in initiating this symbiosis such as the symbiont-encoded 'TCT toxin', which is related to the tracheal cytotoxin of *Bordetella pertussis* (*McFall-Ngai et al., 2013*). A few studies of intracellular insect symbionts have shown that they use type III and type IV secretion systems to establish and maintain their association with their host (reviewed by *Dale and Moran, 2006*; *Snyder and Rio, 2013*). These secretion systems are commonly used by intracellular pathogens to hijack host cell processes, allowing their entry and survival within host cells (e.g., *Hueck, 1998*; *Steele-Mortimer et al., 2002*; *Backert and Meyer, 2006*). An example is the *Sodalis* symbionts of aphids and weevils, which use a type III secretion system for entry to the host cell and are thought to have evolved from pathogens (*Dale et al., 2001*; *Clayton et al., 2012*). The virulence determinants of their pathogenic ancestors might therefore have been co-opted for use in beneficial interactions with their insect hosts.

In contrast to the *Sodalis* symbionts of insects and the *Vibrio* symbionts of squid, the *Bathymodiolus* SOX symbionts are not closely related to any known pathogens. Moreover, because they fall interspersed between free-living SOX bacteria in 16S rRNA phylogenies, they are hypothesized to have evolved multiple times from free-living ancestors (*Figure 2—figure supplement 1*) (*Petersen et al., 2012*). Comparative genomics is a powerful tool for identifying the genomic basis of beneficial and pathogenic interactions, particularly if the symbionts or pathogens have close free-living relatives that do not associate with a host (e.g., *Galagan, 2014*; *Ogier et al., 2014*; *Zuleta et al., 2014*). Genomes of closely related free-living and symbiotic relatives of *Bathymodiolus* SOX symbionts were recently published. Their closest free-living relatives are marine SOX bacteria called SUP05, which are abundant in the world's oceans, particularly in oxygen minimum zones (OMZs) and hydrothermal plumes (*Sunamura et al., 2004*; *Lavik et al., 2009*; *Walsh et al., 2009*; *Anantharaman et al., 2012*; *Wright et al., 2012*; *Petersen and Dubilier, 2014*). The *Bathymodiolus* SOX symbionts and SUP05 bacteria form a monophyletic clade together with the SOX symbionts of vesicomyid clams based on 16S rRNA gene phylogenies (*Figure 2—figure supplement 1*) (*Distel et al., 1995*; *Petersen et al., 2012*). Closed genomes are available for the symbionts of two clam species (*Kuwahara et al., 2007*; *Newton et al., 2007*).

All members of this monophyletic group (the mussel and clam symbionts, and SUP05) share similar core metabolic features. They are all capable of autotrophic growth, and all use reduced sulfur compounds as an energy source (*Newton et al., 2008*; *Walsh et al., 2009*). They can differ in auxiliary metabolic capabilities such as hydrogen oxidation, nitrate reduction, or mixotrophy (*Newton et al., 2008*; *Petersen et al., 2011*; *Anantharaman et al., 2012*; *Murillo et al., 2014*). However, the major difference between these organisms is their lifestyle: SUP05 bacteria are exclusively free-living. The clam symbionts are exclusively host-associated, are vertically transmitted, and have reduced genomes. The *Bathymodiolus* symbionts appear to have adapted to both niches, as they have a host-associated stage and are assumed to also have a free-living stage.

The goal of this study was to identify the genomic basis of host-symbiont interactions in *Bathymodiolus* symbioses. We used high-throughput sequencing and binning techniques to assemble the first essentially complete draft genomes of the SOX symbionts from *Bathymodiolus* mussels. We used comparative genomics of the symbionts' genomes to those of their close free-living and obligate symbiotic relatives to reveal genes potentially involved in *Bathymodiolus* host-symbiont interactions. We used phylogenetics and bioinformatic prediction of horizontally acquired genes to investigate the origins of these genes. Finally, we used transcriptomics and proteomics to determine whether potential host-symbiont interaction genes are being expressed by the symbionts in their host.

## Results

### Draft genome sequences of *Bathymodiolus* symbionts

We sequenced the genomes of the SOX symbionts from three *Bathymodiolus* individuals: two were *Bathymodiolus azoricus* from the Menez Gwen vent field on the northern Mid-Atlantic Ridge (MAR) (*Figure 1*). We refer to these as BazSymA and BazSymB. The third mussel individual was an undescribed *Bathymodiolus* species (BspSym), from the Lilliput hydrothermal vent on the southern MAR (SMAR) (*Figure 1*). Symbiont draft genomes from each individual were almost complete (see 'Materials and methods'). Despite different sequencing and assembly strategies, the draft genomes

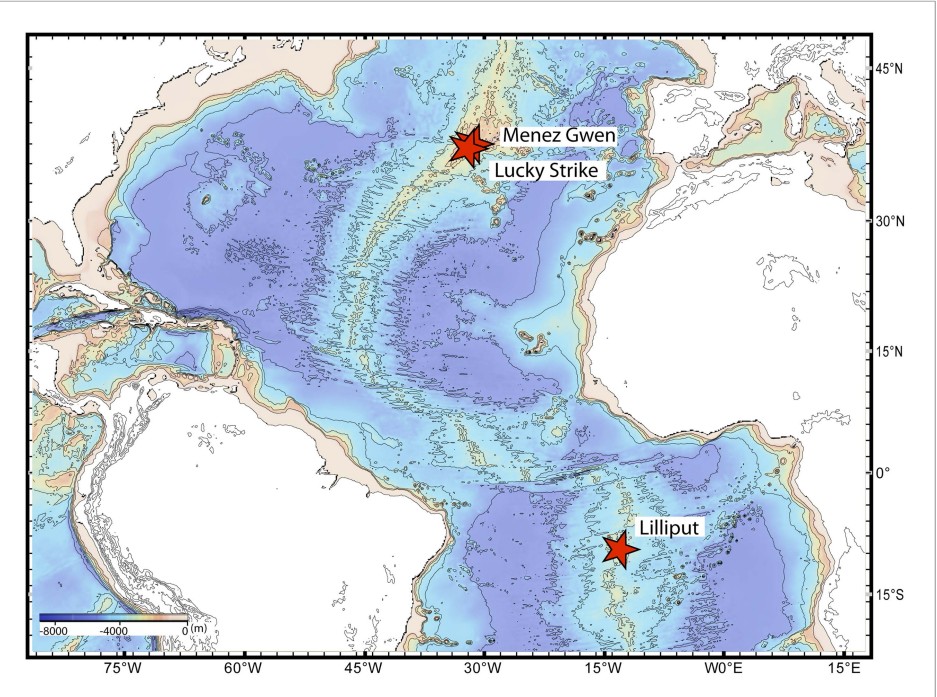

**Figure 1**. Sampling sites. Map showing the sampling sites of *Bathymodiolus* mussels at hydrothermal vents along the Mid-Atlantic Ridge (red stars). *B.* sp. is found at Lilliput (BspSym), *Bathymodiolus azoricus* at Menez Gwen (BazSymA and BazSymB) and Lucky Strike. The details of the sampling sites are described in ***Supplementary file 1E***. The map was produced with GeoMapApp 3.3.

were 90.7–97.7% complete (***Table 1***). The total assembly sizes were between 1.7 and 2.3 Mbp, on 52 to 506 contigs (***Table 1***). Each draft genome contained one copy of the 16S rRNA gene. The BazSymB assembly only contained an 829 bp fragment of the 16S rRNA gene; however, we PCR amplified and sequenced this gene from the DNA used to generate the metagenome. The 16S rRNA genes from the two *B. azoricus* symbionts were 100% identical and were 99.3% identical to BspSym. The core metabolic potential of the *Bathymodiolus* SOX symbionts is described in Appendix 1-Symbiont metabolism. A detailed description of the genomes is beyond the scope of this article and will be published elsewhere.

## General genome comparison

*Bathymodiolus* symbiont genomes were more similar to the free-living SUP05 than to the clam symbionts in terms of size and GC content (***Table 1***). Analysis of codon usage showed that all three *Bathymodiolus* SOX symbiont genomes had a greater proportion of genes that may have been acquired through recent horizontal gene transfer (HGT) compared to the clam symbionts *Candidatus* Vesicomyosocius okutanii, and *Candidatus* Ruthia magnifica (***Table 1***). BspSym, the genome that assembled into the fewest contigs, lacked synteny compared to SUP05 and the clam symbionts, as shown by whole genome alignment (***Figure 2—figure supplement 2***). Genome alignment of BazSymA and BazSymB was not attempted because the assemblies were highly fragmented. The possibility of incorrect genome assembly for BspSym was ruled out for four regions by PCR amplification of sequences spanning the regions without synteny. For confirmation, one PCR product was Sanger-sequenced and found to be identical to the draft genome assembly of BspSym. These regions without genome synteny therefore most likely represent true genome reshuffling in *Bathymodiolus* symbionts.

The *Bathymodiolus* symbiont genomes had more mobile elements compared to their closest relatives (***Supplementary file 1A***). *Bathymodiolus* symbionts had between 13 and 23 transposases and three to five integrases. SUP05 had 14 transposases and one integrase. We did not find any

**Table 1.** Overview of the genomes compared in this study: SOX symbiont of *B. sp*, two individual SOX symbionts of *B. azoricus*, SOX symbiont *Candidatus* Vesicomyosocious okutanii, SOX symbiont of *Calyptogena magnifica* (*Candidatus* Ruthia magnifica), and free-living SUP05

| Genome | Collection site | Contigs | GC content (%) | Length/Span (Mbp)† | Number of CDSs | HGT | Estimated completeness‡ | Coverage§ | Separation method# | References |
|---|---|---|---|---|---|---|---|---|---|---|
| *B. sp* symbiont (BspSym) | Lilliput | 52 | 38.23 | 1.8/2.3 | 2225 | 33% | 95.39% | 22X | Filtration | *Petersen et al., 2011*, this study |
| *B. azoricus* symbiont (BazSymB) | Menez Gwen | 239 | 38.20 | 1.5/1.7 | 1802 | 30% | 90.60% | 8X | Gradient centrifugation/binning | This study |
| *B. azoricus* symbiont (BazSymA)* | Menez Gwen | 506 | 37.58 | 1.85/1.85 | 2008 | 35% | 97.70% | 59X | Binning | This study |
| Ca. V. okutanii | Sagami Bay | 1 | 31.59 | 1.0/1.0 | 980 | 26% | 93.58% | – | Whole genome assembly | *Kuwahara et al., 2007* |
| Ca. R. magnifica | East Pacific Rise, 9°N | 1 | 34.03 | 1.2/1.2 | 1210 | 23% | 94.84% | – | Whole genome assembly | *Newton et al., 2007* |
| SUP05 | Saanich Inlet | 97 | 39.29 | 1.4/2.5 | 1586 | 30% | 85.76% | – | Binning | *Walsh et al., 2009* |

SOX, sulfur-oxidizing.

*SOX symbiont sequences recovered from metagenome of adductor muscle.

HGT = Genes that potentially originated from horizontal gene transfer.

†Length is the total length of sequence information on contigs without Ns, and span is the entire length of scaffold assembly including Ns.

‡The completeness of the genome was estimated with CheckM using a set of lineage-specific genes for proteobacteria (*Parks et al., 2015*).

§Median coverage.

#Separation method indicates the experimental separation of symbionts from host tissue and co-occurring symbionts (filtration or gradient centrifugation), or the in silico separation of genomic information from hosts and co-occurring bacteria (binning).

transposases or integrases in the clam symbiont genomes. The *Bathymodiolus* symbionts were highly enriched in restriction-modification system genes (between 10 and 22 genes), whereas SUP05 only had one, and the clam symbionts had none. This large difference raises the possibility that restriction-modification systems are involved in genome reshuffling in the *Bathymodiolus* symbionts.

## Gene-based comparison reveals toxin-related genes specific to *Bathymodiolus* symbionts

Between 2.3 and 7.6% of the genes found only in the *Bathymodiolus* SOX symbionts but not in the clam symbionts and SUP05, genomes were annotated as toxin or virulence genes (*Figure 2*). Most were related to genes from one of three toxin classes: (1) the RTX (repeats in toxins) toxins, (2) MARTX (multifunctional autoprocessing RTX toxins), a sub-group of RTX toxins, and (3) YD repeat toxins (also called *rhs* genes as they were initially described as '*recombination hotspots*'). Representatives from all three toxin-related genes (TRGs) classes were found in each of the three-draft genomes, except for RTX, which were not found in BazSymA. The number of genes from each class varied between the three genomes. We found the largest number of TRGs in the genome with the fewest contigs, BspSym, which had at least 33 YD repeat genes, eight RTX genes, and 19 MARTX-like genes. In the BazSymB genome, 14 YD repeat genes, two RTX genes, and up to 10 MARTX genes were found. BazSymA had 16 YD repeat genes, and one MARTX (*Figure 2*, *Supplementary file 1B*). This indicates that these toxin-related classes are common to the SOX symbionts of both *B.* sp. and *B. azoricus*. In the BspSym genome, which assembled into the largest contigs, 22 out of 88 TRGs were found directly upstream or downstream of mobile elements.

Toxin genes are known to have unusually high substitution rates due to an 'evolutionary arms race' with their targets (*Jackson et al., 2009*; *Linhartová et al., 2010*). Accordingly, many of the *Bathymodiolus* symbiont TRGs were highly variable between the symbionts of *B. azoricus* and *B.* sp., but also between the symbionts from the two *B. azoricus* individuals, and even between different copies within one genome (see 'Variability of TRGs within *Bathymodiolus* SOX symbiont populations'). We therefore searched for homologs of the TRGs in the clam symbionts and SUP05 with a lower BSR of up to 0.25, but even with this reduced stringency, no hits were found (see 'Materials and methods').

We ruled out the possibility that the TRGs were not found in SUP05 draft genome because of its incompleteness (~89% complete), by searching for homologs of the *Bathymodiolus* symbionts TRGs in unbinned metagenomes and metatranscriptomes from hydrothermal plumes and OMZs that are enriched in SUP05. If free-living SUP05 also encoded these TRGs, we would expect to find them regularly in SUP05-enriched metagenomes and metatranscriptomes. Instead, no hits were found in four out of these six data sets. In a metagenome from the Lost City hydrothermal vent, we found one weak hit to a YD repeat gene (31% similarity). In a metagenome from the Guaymas Basin hydrothermal vent, we found one weak hit to an RTX gene (34% similarity). However, both of these metagenomes were from sites colonized by either *Bathymodiolus* mussels (Lost City), or *Riftia pachyptila* tubeworms (Guaymas Basin), whose symbionts also encode a hemolysin gene of the RTX class (*Gardebrecht et al., 2012*). These rare hits might therefore come from contamination by symbionts in the environment (*Harmer et al., 2008*). Considering the almost complete absence of genes similar to the TRGs of *Bathymodiolus* in SUP05-enriched next-generation sequence data sets, we conclude that these genes are specific to the *Bathymodiolus* SOX symbionts and are not found in their close symbiotic or free-living relatives.

## Relationships to other toxins

Most symbiont TRGs were so divergent that they could not be confidently aligned. One exception was the YD repeat genes, a few of which contained a conserved repeat region. We reconstructed the phylogeny of this conserved region. YD repeat sequences from the symbionts of both *Bathymodiolus* species formed a distinct cluster, distant from all other sequences in public databases (*Figure 3*, *Figure 3—figure supplement 1*). The *Bathymodiolus* symbiont sequences did not cluster according to their host species, but instead were intermixed, suggesting gene duplication events prior to the divergence of the *B. azoricus* and *B.* sp. symbiont lineages. The *Bathymodiolus* symbiont sequences fell into a cluster that contained mostly pathogenic bacteria such as *Yersinia pestis* and *Burkholderia pseudomallei*. This cluster was well supported by Bayesian analysis (0.91 posterior probability). This cluster also contained a number of beneficial symbionts such as *B. rhizoxinica*, which is an intracellular

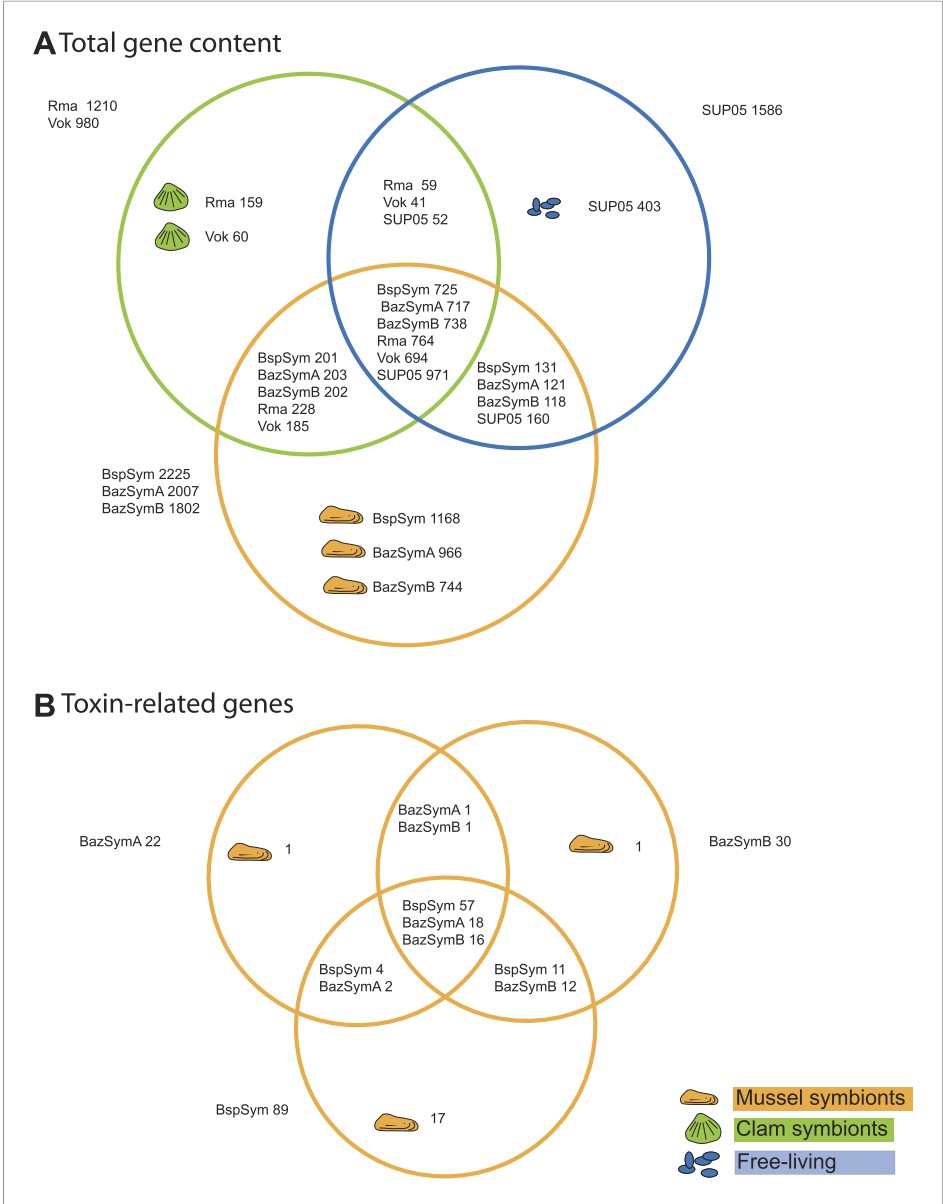

**Figure 2**. Genes shared between the *Bathymodiolus* and vesicomyid SOX symbionts and free-living SUP05. Protein-coding sequences from the *Bathymodiolus* sulfur-oxidizing (SOX) symbiont were compared to the clam symbiont genomes and to the SUP05 metagenome from *Walsh et al. (2009)* with BLAST score ratios (BSR). (**A**) Venn diagram of the shared and unique gene content in the clam symbionts, mussel symbionts, and SUP05 bacteria. Predicted protein sequences of each mussel symbiont were compared to a combined data set of the clam symbionts (Rma and Vok) and SUP05. Similarly, protein sequences of each clam symbiont were compared to a combined data set of mussel symbionts (BspSym, BazSymB, and BazSymA). Depending on the reference genome, the number of shared genes varies slightly and possibly reflects the presence of paralogous genes and redundant sequence information in these draft genomes. Abbreviations are explained in detail in *Table 1*. The BLAST score ratio (BSR) threshold was 0.4. (**B**) Venn diagram of mussel symbiont toxin-related genes (TRGs), calculated with a BSR threshold of 0.2.

The following figure supplements are available for figure 2:

**Figure supplement 1**. Maximum likelihood 16S rRNA phylogeny of the close relatives of the *Bathymodiolus* SOX symbionts.

**Figure supplement 2**. Whole genome alignment.

*Figure 2. Continued*

**Figure supplement 3**. Metabolic reconstruction of the Bathymodiolus symbiont.

symbiont of the fungus *Rhizopus*, and the *Photorhabdus* and *Xenorhabdus* symbionts of soil nematodes (***Waterfield et al., 2001***; ***Goodrich-Blair and Clarke, 2007***; ***Moebius et al., 2014***).

To overcome the difficulties in aligning these highly divergent TRGs, we constructed gene sequence similarity networks based on BLAST to depict relationships among and between the symbiont TRGs, and those in public databases. This analysis revealed distinct sequence clusters that contained genes with >25% similarity over at least half of the length of the gene (***Figure 4***). If a cluster

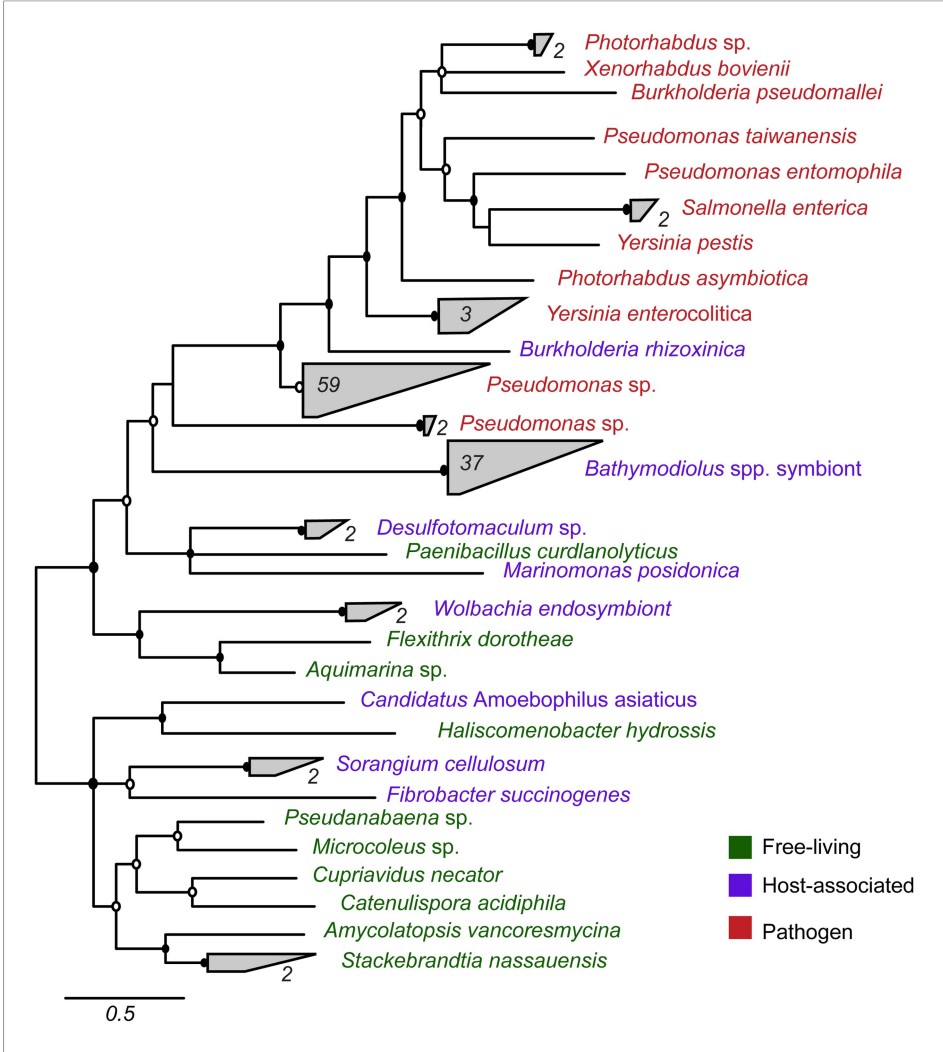

**Figure 3**. Phylogeny of YD repeat-containing proteins. The tree is a consensus of bayesian and maximum likelihood analyses, result of an alignment of 536 amino acids. Black circles represent branches with posterior probability >0.8 and bootstrap value >80. White circles represent branches with either posterior probability >0.8 or bootstrap value >80. The number of sequences per collapsed group is shown next to the gray bloks. Purple: organism found in intestinal microflora or in close association with another organism; green: free-living; red: pathogen.

The following figure supplement is available for figure 3:

**Figure supplement 1**. Consensus of bayesian and maximum likelihood phylogeny of YD proteins with identifiers.

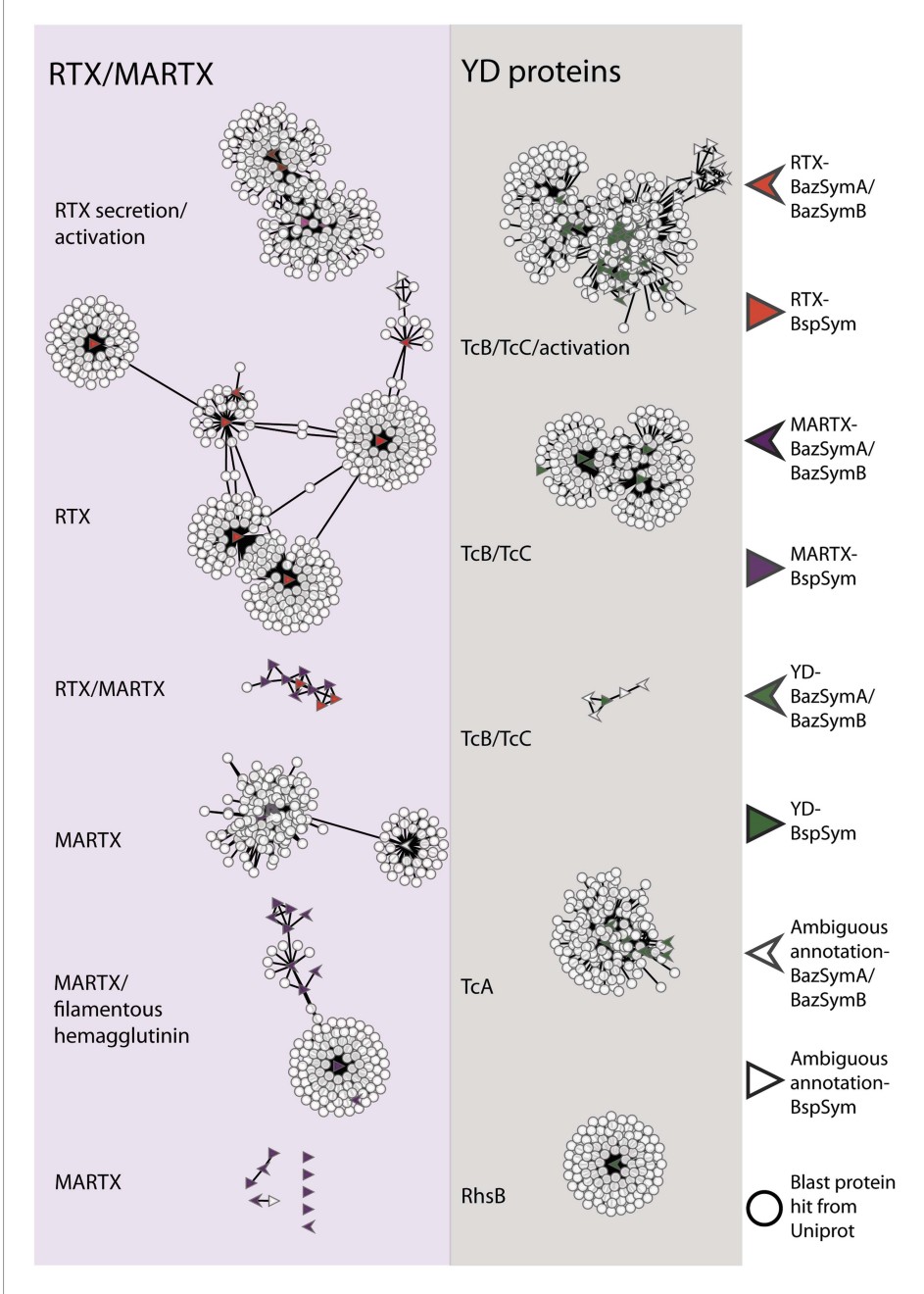

**Figure 4**. Protein similarity network of toxin-related proteins in the *Bathymodiolus* symbionts. Each node corresponds to a protein sequence and the links between nodes represent BLAST hits. The length of the edges is inversely proportional to the sequence similarity. Protein clusters containing RTX or multifunctional autoprocessing RTX (MARTX) proteins are shown in the red panel on the left, and sequence clusters containing YD repeats are shown in the gray panel on the right. Arrowheads are proteins from *B. azoricus* symbionts, and triangles are proteins from *B.* sp. symbionts. The symbols are colored in green if they were identified in the Bathymodiolus symbionts as YD repeat-containing genes, red if they were identified as RTX genes, and purple for MARTX genes. Some protein sequences were similar to the TRGs but not annotated as such as these are partial genes that did not have any conserved domain. If the clusters contained mostly genes with a particular annotation, we named the clusters after these annotations, for example, cluster 'TcB/TcC' contained proteins annotated as TcB or TcC.

*Figure 4. continued on next page*

*Figure 4. Continued*

The following figure supplements are available for figure 4:

**Figure supplement 1**. Network of toxin-related proteins in the *Bathymodiolus* symbionts with BLAST hits from *Vibrio*, *Photorhabdus*, *Xenorhabdus*, and *Pseudomonas* highlighted.

**Figure supplement 2**. Genomic architecture of MARTX regions.

contained at least one gene that was similar to at least one other gene in another cluster (similarity cut-offs as above), then these clusters were joined to create a larger network. They were also joined if both clusters contained genes that had similarity to another gene in the database (i.e., if they could be joined by at most two steps). This allowed us to identify distinct sub-groups within the three toxin-related classes, and to identify toxin sequences from public databases that were most similar to the *Bathymodiolus* symbiont genes.

TRGs from the *Bathymodiolus* symbionts clustered together with toxin and TRGs from phylogenetically diverse organisms including characterized toxins of gammaproteobacterial *Vibrio* and *Pseudomonas*, and TRGs of the gammaproteobacteria *Shewanella*, the actinobacterial *Rhodococcus,* the cyanobacterium *Trichodesmium*, and the firmicute *Caldicellulosiruptor* (see e.g., *Figure 4—figure supplement 1*). The RTX genes clustered into two separate networks, one that had similarity to RTX secretion and activation genes, and one that had similarity to RTX toxins. Five distinct networks contained MARTX genes. One of these included genes from the symbiont MARTX1 cluster, and genes from other organisms that were annotated as MARTX or filamentous hemagglutinin. One network contained some genes that we classified as RTX and some we classified as MARTX, reflecting their shared features such as the RTX repeats. Eight MARTX genes had no significant hits to any other gene in public databases.

The YD repeat genes formed five distinct networks. Sequences from the first three had structural similarity to TcB and TcC, two subunits of the ABC toxins of *Photorhabdus* and *Xenorhabdus,* the beneficial symbionts of entomopathogenic nematodes. The B and C subunits form a cage-like structure that encapsulates the toxic domain (an adenosine diphosphate (ADP) ribosylation domain, located at the C terminus of the C subunit). The A subunit forms a syringe-like structure, which delivers the toxin to the insect cell (*Meusch et al., 2014*). The genes in the fourth YD network had structural similarity to TcA genes that encode the syringe-like A subunit. The fifth YD network had similarity to genes annotated as RhsB, which was shown to play a role in bacteria–bacteria competition in *Escherichia coli* (*Poole et al., 2011*).

## MARTX and YD repeat genes are enriched in the genomes of host-associated bacteria

The *Bathymodiolus* symbiont genomes encoded more YD repeat and MARTX genes than any other genome that we compared them to (*Figure 5*, *Figure 5—figure supplements 1–3*). This is remarkable considering the relatively small size of their genomes, and the fact that they are still incomplete. A few published genomes encoded more RTX genes, but these were much larger (>5 Mbp) (*Figure 5—figure supplement 3*).

The vast majority of RTX, MARTX, and YD repeat proteins have not been functionally characterized. The few proteins whose function has been studied in detail are from bacteria that are known pathogens or cultured strains that can form biofilms. Because of this, it is generally assumed that RTX, MARTX, and YD repeat proteins function in host-microbe interactions, in microbe–microbe antagonism, or in biofilm formation, but this has not been extensively tested. To further investigate the functional role of the TRGs encoded by the *Bathymodiolus* symbionts, we tested whether similar genes are more likely to be found in bacteria that live in a particular niche (extracellular host-associated, intracellular host-associated, or free-living), or that express a particular phenotype (pathogenesis or biofilm formation).

First, we used the Kruskal–Wallis one-way analysis of variance to determine whether the distribution of the three TRGs classes differed significantly (I) between biofilm-forming vs non-biofilm-forming bacteria, (II) between pathogenic and non-pathogenic bacteria, and (III) between free-

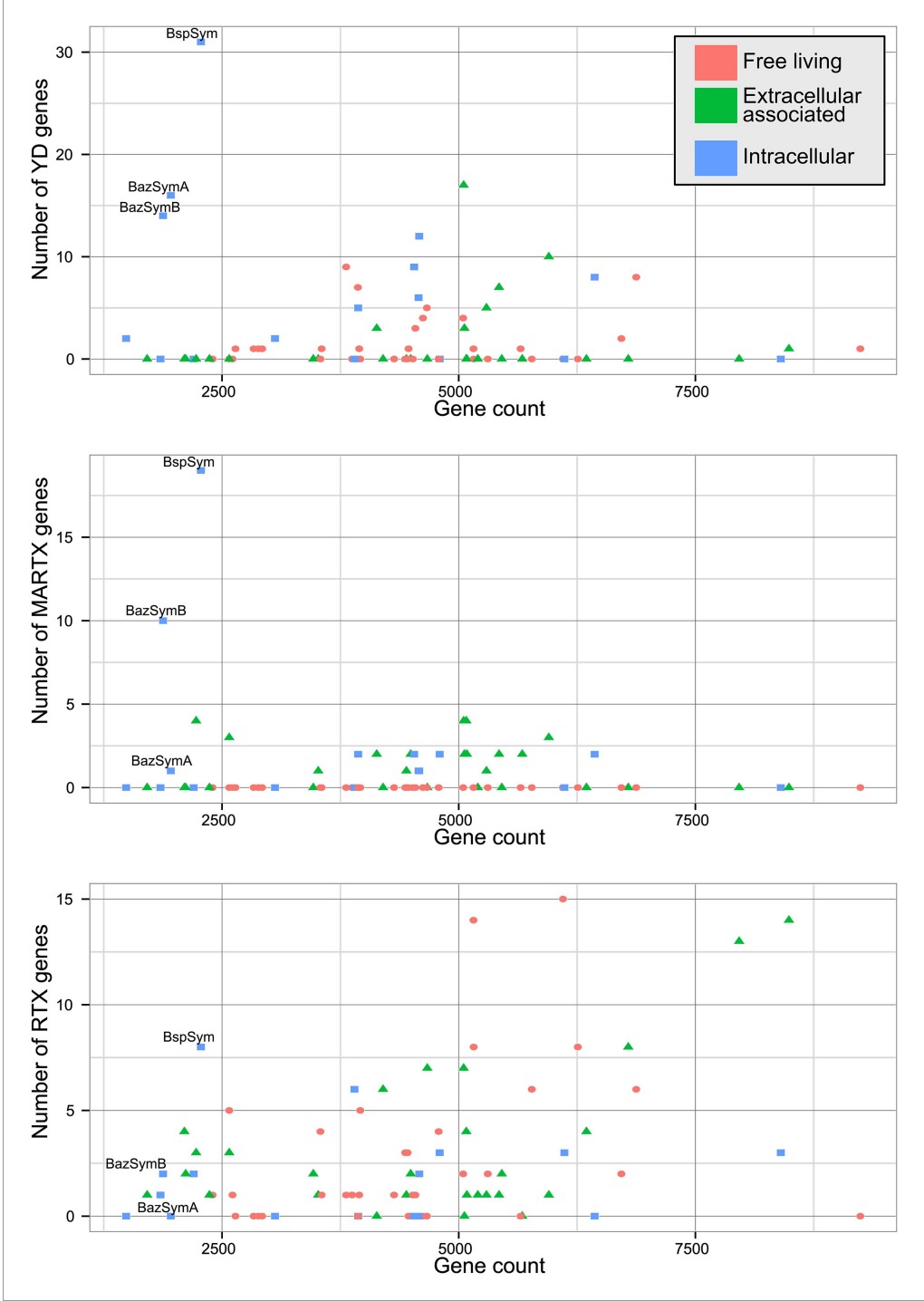

**Figure 5**. Distribution of the three major TRGs classes according to lifestyle. Each dot represents one sequenced genome. The sum of TRGs is on the Y axis, and the total number of genes predicted in each genome is on the X axis. Free-living bacteria are shown in red, host-associated bacteria that live outside of host cells are in green, and host-associated bacteria that can live inside host cells are shown in blue. The positions of the Bathymodiolus SOX symbionts are indicated. A detailed overview of all organisms that had similar TRGs to the SOX symbiont with the number of TRGs is shown in *Supplementary file 1B*.

The following figure supplements are available for figure 5:

**Figure supplement 1**. YD genes per genome, normalized to the total gene count.

*Figure 5. continued on next page*

*Figure 5. Continued*

**Figure supplement 2**. MARTX genes per genome, normalized to the total gene count.

**Figure supplement 3**. RTX genes per genome, normalized to the total gene count.

living bacteria, host-associated intracellular bacteria, and host-associated extracellular bacteria, without considering whether the bacteria were pathogenic. There was no significant enrichment of any TRG category in bacteria known to form biofilms vs those that do not (*Table 2*). One class, MARTX was significantly enriched in the genomes of pathogenic vs non-pathogenic bacteria (p-value = 0.007, Kruskal–Wallis test). There was also a significant bias in the distribution of genes encoding MARTX and YD repeat genes in bacteria according to their lifestyle (extracellular host-associated, intracellular host-associated, or free-living).

When three categories are tested, such as in (III) above, the Kruskal–Wallis test does not identify which category the bias is associated with. To tease apart which of these three niche categories was most enriched in TRGs, we did Mann–Whitney–Wilcoxon tests (*Table 3*). These showed that both YD repeat and MARTX genes were enriched in the genomes of host-associated microbes (YD repeat: p-value = 0.026, MARTX: p-values = $2.125e^{-6}$, $1.618e^{-6}$, Mann–Whitney–Wilcoxon test). While MARTX genes were enriched in host-associated bacteria regardless of their location, YD repeat genes were only significantly enriched in intracellular bacteria. In contrast to YD repeat and MARTX genes, RTX did not show any enrichment in the three defined categories. RTX are therefore widely distributed among bacteria and are just as likely to be found in free-living and host-associated bacteria (Appendix 2).

Bacteria that are closely related often have similar genomic and physiological features. However, toxin genes are commonly gained through HGT, which may weaken the phylogenetic signal in their distribution patterns (reviewed by *Dobrindt et al., 2004*; *Gogarten and Townsend, 2005*). To tease apart the possible phylogenetic influence on the TRGs distribution, we used Permanova to test whether any of the three classes was enriched in particular phylogenetic groups at the class, order, and family levels. Only RTX genes were significantly enriched, and only at the order level (p-value = 0.0159) (*Supplementary file 1C*). Therefore, phylogeny is not the main driver in the toxin-related distribution of YD repeats genes and MARTX.

## Variability of TRGs within *Bathymodiolus* SOX symbiont populations

Toxin genes often have unusually high substitution rates, making them highly variable compared to non-toxin genes (e.g., *Ohno et al., 1997*; *Davies et al., 2002*). We compared the substitution rates of all genes in the *Bathymodiolus* SOX symbiont genomes within the population of symbionts associated with each mussel species. This was done for each of the two species, *B. azoricus* and *B.* sp. by mapping transcriptome reads from three *Bathymodiolus* individuals to the draft genomes of their respective symbionts (see below for transcriptomes). We calculated the number of single nucleotide polymorphisms (SNPs) per kb per gene. The number of SNPs in most of the TRGs was not significantly higher than the genome-wide average (*Figure 6*). However, we found 22 TRGs that did have

**Table 2**. p-values obtained with Kruskal–Wallis rank sum test

|  | B/NB df = 1 | P/NP df = 1 | Ext/Int/FL df = 2 |
|---|---|---|---|
| YD | 0.097 | 0.5217 | 0.010* |
| RTX | 0.715 | 0.793 | 0.308 |
| MARTX | 0.773 | 0.007* | $3.21e^{-06}$* |

The three main lifestyle categories were tested against each toxin-related class. Number of TRGs per genome was normalized to the total gene count.

FL = free-living, Ext = extracellular host-associated, Int = intracellular host-associated, P = pathogen, NP = non-pathogen, B = found in biofilms, NB = not found in biofilms, df = degrees of freedom, TRG, toxin-related gene, MARTX, multifunctional autoprocessing RTX.

*p-value was considered to be significant (p < 0.05).

**Table 3**. p-values obtained with Mann–Whitney–Wilcoxon test for enrichment of YD and MARTX genes similar to those from the SOX symbiont

|  | FL/Ext | FL/Int | Ext/Int |
|---|---|---|---|
| YD | 0.129 | 0.026 | 0.006* |
| MARTX | $2.125e^{-06}$* | $1.618e^{-06}$* | 0.751 |

FL = free-living, Ext = extracellular host-associated, Int = intracellular host-associated, MARTX, multifunctional autoprocessing RTX, SOX, sulfur-oxidizing. *p-value was considered to be significant ($p < 0.05$).

significantly more SNPs than most of the other genes in the genomes. Among the 22 highly variable genes, we found representatives of each TRG class: YD repeats, RTX, and MARTX (*Figure 6*).

## Expression of TRGs

Transcriptome sequencing revealed that all predicted TRGs of the SOX symbionts in *B. azoricus* and *B.* sp gills were expressed. Reads mapping to TRGs accounted for 0.67–1.71% of mRNA in *B. azoricus* symbionts and 0.58–3.14% in *B.* sp. symbionts. All TRGs were found in the transcriptomes of at least one of the three individuals that we sequenced per species (*Supplementary file 1D*). The expression levels of some genes from the RTX, MARTX, and YD repeats classes were in some cases higher than the expression of the essential Calvin cycle gene ribulose bisphosphate carboxylase/oxidase (RuBisCO), which

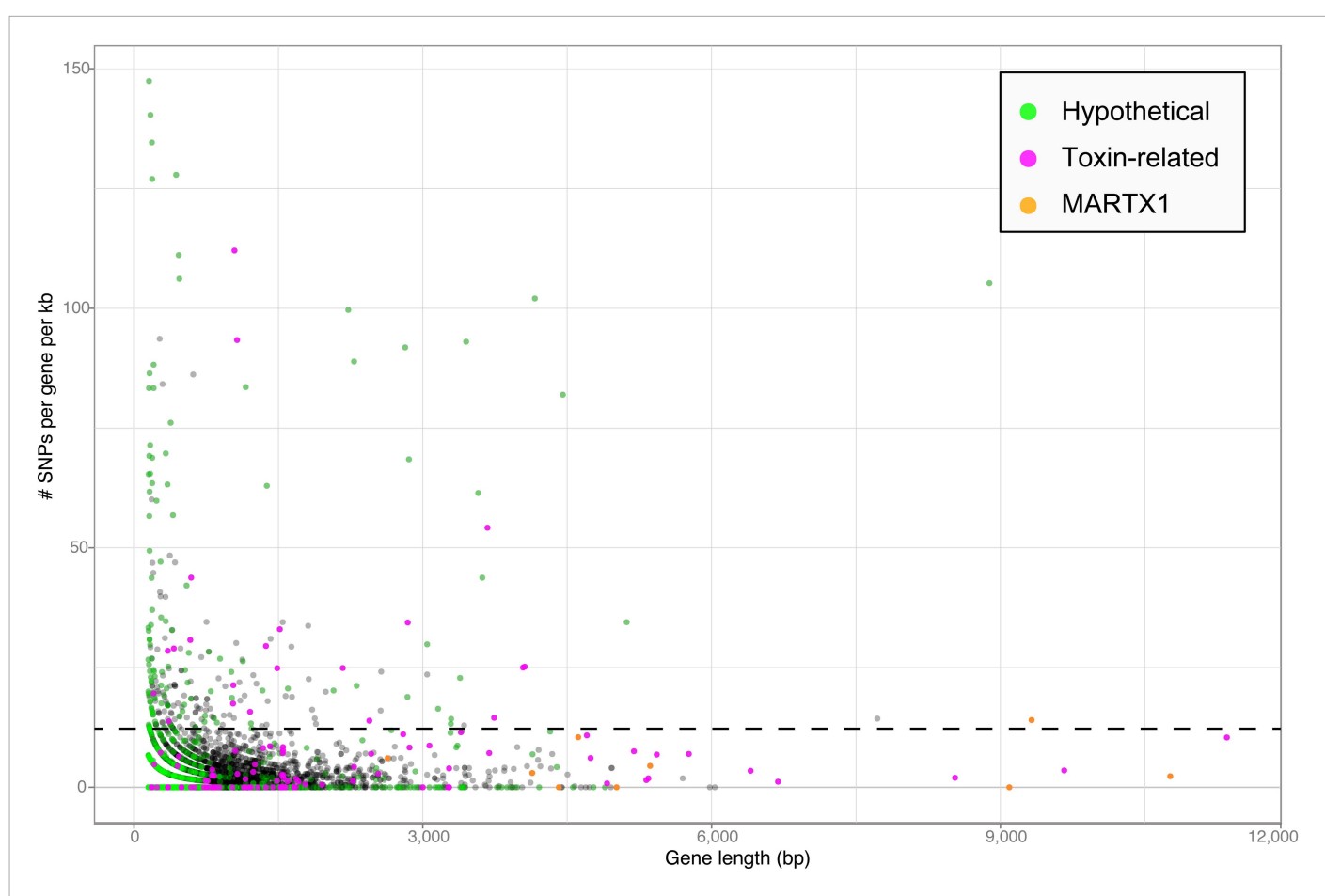

**Figure 6**. Single nucleotide polymorphisms per gene. The number of single nucleotide polymorphisms (SNPs) per gene was normalized according to the length minus regions of unknown sequence for genes containing N's. Genes smaller than 150 bp were not considered. The dotted line represents the median plus one standard deviation of the number of SNPs per gene per kb.

The following source data is available for figure 6:

**Source data 1**. Variability in TRGs encoded by the *Bathymodiolus* SOX symbionts.

accounted for 0.03–0.5% of mRNA in *B. azoricus* symbionts and 0.11–1.04% in *B*. sp. symbionts. We could also confirm expression of some TRGs at the protein level in *B. azoricus* mussels (samples of *B*. sp. were not available for proteomics). We analyzed (I) gradient centrifugation fractions that were enriched in the *Bathymodiolus* SOX symbiont and (II) whole gill tissue. Some of the characterized toxin proteins are associated with membranes. We therefore also analyzed a fraction enriched in membrane proteins. In these proteomes, we found 12 YD repeat proteins and one RTX. We also found a number of toxin-associated proteins such as one RTX activator and two RTX toxin transporters (*Table 4*). Nine of the 12 identified YD repeat proteins were present in the membrane proteome. Our method allowed us to identify symbiont-encoded proteins that were found in higher relative abundance in the whole gill fraction compared to the symbiont-enriched fraction (see Appendix 4). These proteins are potentially exported to the host tissue, indicating that they play a role in direct host-symbiont interactions. One RTX activator and one YD repeat protein were significantly enriched in the host tissue (*Table 4*).

## Discussion

### Origin of TRGs in *Bathymodiolus* symbionts

Two scenarios could explain the origin of the large complement of TRGs in the *Bathymodiolus* symbionts and not in their close relatives: firstly, the TRGs could have been in the genome of their last

**Table 4**. Toxin-related proteins found in the proteome of the SOX symbiont from *B. azoricus*

| Proteome | Identifier | Category | Annotation | Molecular weight (kDa) | Max. number of unique peptides* |
|---|---|---|---|---|---|
| SM | Host_EST_000107 | YD | IPR006530IYD repeat | 43 | 11 |
| SM | Host_EST_000115 | YD | IPR006530IYD repeat | 42 | 12 |
| N | Host_EST_000248 | YD | IPR006530IYD repeat | 37 | 7 |
| M | Host_EST_002123 | YD | IPR006530IYD repeat | 24 | 5 |
| SM† | Thio_BAZ_1943_contig360420_0 | RTX (activator) | Hemolysin-activating lysine-acyltransferase (Hemolysin C) | 19 | 3 |
| SM | Tox_BAZ_119_contig00027_0 | YD | RHS repeat-associated core domain-containing protein | 202 | 17 |
| SM | Tox_BAZ_120_contig00027_1 | YD | Virulence plasmid 28.1 kDa A protein | 62 | 11 |
| SM | Tox_BAZ_1734_contig02141_2 | RTX (transporter) | Secretion protein HlyD family protein | 43 | 10 |
| SM | Tox_BAZ_2494_contig00030_0 | YD | Virulence plasmid 28.1 kDa A protein | 183 | 33 |
| M | Tox_BAZ_3202_scaffold00038_7 | RTX | Hemolysins and related proteins containing CBS domains | 35 | 2 |
| SM | Tox_BAZ_525_contig104979_0 | YD | Virulence plasmid 28.1 kDa A protein | 52 | 2 |
| S† | ToxAzor_892893 | YD | Rhs | 114 | 2 |
| SM | ToxSMAR_1260BAT01109 | YD | [weak similarity to] Toxin complex/plasmid virulence protein | 321 | 8 |
| N | ToxSMAR_2052BAT01788, Thio_BAZ_1733_contig02141_1 or Thio_BAZ_2580_scaffold00010_8 | RTX (transporter) | Toxin secretion ATP-binding protein | 79 | 5 |
| S | ToxSMAR893-894 | YD | Rhs family protein | 103 | 1 |
| SM | ToxAzor_890891 | YD | Rhs family protein | 67 | 6 |

S = soluble proteome, M = membrane-enriched proteome, SM = found in both proteomes, SOX, sulfur-oxidizing.
*The highest number of unique peptides detected in one sample.
†Proteins that are potentially exported by the symbiont to the host gill tissue.

common ancestor but were all subsequently lost in both the clam symbionts and SUP05. Alternatively, the *Bathymodiolus* symbionts could have acquired these genes via HGT after their divergence from the clam symbionts and SUP05. Toxins are often found in 'genomic islands' that have been acquired through HGT (reviewed by *Lindsay et al., 1998*; *Ochman et al., 2000*; *Dobrindt et al., 2004*; *Soucy et al., 2015*). Several observations point towards HGT of the TRGs into the *Bathymodiolus* symbionts rather than their loss by the clam symbionts and SUP05. Firstly, 63–68% of the TRGs could be identified as potentially horizontally acquired based on codon usage analysis, in contrast to 30–35% predicted for all coding sequences (*Table 1*, *Figure 6—source data 1*). This means that their transfer was relatively recent, as the codon usage of these genes has not yet adapted to one typical of the symbiont genomes. Secondly, the content of the *Bathymodiolus* symbiont genomes attest to the major role that HGT has played in their evolution. They are enriched in mobile elements such as transposases and restriction-modification systems compared to their closest relatives (*Supplementary file 1A*). The lack of synteny we observed in the symbiont genomes is consistent with the presence of mobile elements and major HGT events (*Kobayashi, 2001*; *Rocha and Danchin, 2002*; *Achaz et al., 2003*). Thirdly, the TRGs from the *Bathymodiolus* symbionts are similar to genes from distantly related bacteria. Finally, mobile elements were regularly found directly upstream or downstream of the TRGs in the *Bathymodiolus* symbiont genomes. The linkage of mobile elements with some of these genes could explain the mechanism of their transfer into the *Bathymodiolus* symbiont genomes. This could also explain why each genome contained multiple copies of TRGs, as mobile elements are also prone to duplication (*Reams and Neidle, 2004*). Considering these observations, and the absence of these TRGs in their close relatives, we consider it most likely that they were acquired through HGT.

## Are some of the TRGs used for direct beneficial interactions between hosts and symbionts?

Genes from the second major class of TRGs, MARTX, that were similar to those from *Bathymodiolus* symbiont, were significantly enriched in beneficial and pathogenic host-associated bacteria. One of the regions encoding MARTX genes from the *Bathymodiolus* SOX symbiont has a domain structure similar to the filamentous hemagglutinin FhaB from *B. pertussis* and *Bordetella bronchiseptica* (MARTX1, see Appendix 3). FhaB is involved in attachment of *Bordetella* to their human host and suppression of the immune response (reviewed by *Melvin et al., 2014*). *B. bronchiseptica* has two distinct phenotypic stages: an infective stage, where *fhaB* is upregulated, and a non-infective stage, where *fhaB* is downregulated. The non-infective stage is necessary for its survival in the environment outside of the host. The lifestyle of the *Bathymodiolus* symbiont has striking similarities to *B. bronchiseptica*. The symbionts must also survive in the environment to be transmitted from one host generation to the next. Our transcriptomes showed that these MARTX genes are expressed by *Bathymodiolus* symbionts within the host tissue. Unfortunately, we do not have samples to test whether they are downregulated in environmental symbiont stages. If so, it may have a similar function to its homologs in pathogenic *Bordetella*. MARTX-like genes also mediate cell–cell attachment in the symbiotic bacterial consortium '*Chlorochromatium aggregatum*' (*Vogl et al., 2008*; *Liu et al., 2013*).

## Are some of the YD repeat proteins used for antagonistic bacteria–bacteria interactions?

Like the MARTX, YD repeat proteins were also significantly more enriched in host-associated compared to free-living bacteria. Some characterized YD repeat proteins function in competition between closely related bacterial strains (*Waterfield et al., 2001*; *Kung et al., 2012*; *Koskiniemi et al., 2013*). In its intracellular niche, why would the *Bathymodiolus* symbiont need to compete against other bacteria? Multiple strains of SOX symbionts can co-occur in *Bathymodiolus* mussels (*Won et al., 2003a*; *DeChaine et al., 2006*; *Duperron et al., 2008*). These may compete with each other for nutrients and energy, or for space within host cells. Bacteria that express toxins to inhibit their close relatives need an immunity protein to protect them from each toxin (*Zhang et al., 2012*; *Benz and Meinhart, 2014*). These immunity proteins are encoded immediately downstream or upstream of the toxin protein. The toxin-immunity pair is usually linked to genes encoding a type VI secretion system, which is the mechanism of toxin delivery for all so far described YD proteins involved in bacteria–bacteria competition (*Zhang et al., 2012*; *Benz and Meinhart, 2014*). None of the YD repeat proteins in the *Bathymodiolus* SOX symbiont genome was found in an operon with an

identifiable immunity protein, and no type VI secretion system gene was found in any of our draft genomes. The arrangement of YD repeat genes in the *Bathymodiolus* symbiont genomes and the lack of genes encoding type VI secretion systems are therefore inconsistent with a role in competition between closely related symbiont strains.

*Bathymodiolus* mussels are also infected by bacterial intranuclear pathogens called *Candidatus* Endonucleobacter bathymodioli, which are related to the genus *Endozoicomonas* (*Zielinski et al., 2009*). *Ca.* E. bathymodioli invades the mussel cell nuclei where it multiplies, eventually bursting the infected cell. Intranuclear bacteria are never found in the nuclei of symbiont-containing cells, which led *Zielinski et al. (2009)* to hypothesize that the symbionts can protect their host cells against infection. Consistent with this hypothesis, growth inhibition assays showed that *B. azoricus* gill tissue homogenates inhibit the growth of a broader spectrum of pathogens compared to the symbiont-free foot tissues (*Bettencourt et al., 2007*). The mechanisms of protection against cultured bacterial pathogens and *Ca.* E. bathymodioli are unknown. There is no evidence from their genomes that the SOX symbionts of *B.* sp. and *B. azoricus* produce antibiotics, but they do have genes for bacteriocin production that were expressed in the six transcriptomes analyzed in this study. Expression of some of the TRGs discovered here, for example, those related to toxins involved in bacteria–bacteria competition, or the production of bacteriocins by symbionts could explain the absence of intranuclear bacteria from symbiont-hosting cells.

## Are some of the TRGs used for protection against eukaryotic parasites?

Symbionts of other marine invertebrate hosts are able to recognize, enter, and survive within host cells without a large number of TRGs. For example, the genomes of the SOX symbionts of hydrothermal vent *Riftia* tubeworms and the heterotrophic symbionts of whale-fall *Osedax* worms are virtually complete, but they only contain one or a few RTX genes, and no YD repeat genes (*Robidart et al., 2008*; *Gardebrecht et al., 2012*; *Goffredi et al., 2014*). There is overwhelming evidence that SOX symbionts are beneficial for their *Bathymodiolus* mussel hosts (Appendix 1). It is therefore highly unlikely that the *Bathymodiolus* SOX symbionts are pathogens that have been mistaken for beneficial symbionts. The remarkably large number of TRGs in the *Bathymodiolus* SOX symbionts bears striking similarity to the arsenal of toxins encoded by *Candidatus* Hamiltonella defensa, which is a facultative symbiont of aphids (*Oliver et al., 2003*, *2005*). Both symbionts encode multiple copies of RTX and YD repeat proteins (*Degnan and Moran, 2008*). *Ca.* Hamiltonella defensa is a defensive symbiont that protects its host from attack by parasitic wasps (*Oliver et al., 2003*). Its protective effect is linked to its complement of RTX and YD repeat toxins (*Degnan and Moran, 2008*; *Oliver et al., 2010*). Based on our phylogeny of YD repeats, those from the *Bathymodiolus* symbionts cluster together with sequences from both pathogens and beneficial symbionts. Of the beneficial symbionts in this cluster, all except the *Bathymodiolus* symbionts have been shown to produce exotoxins that damage the organisms their host parasitizes, either a plant in the case of *B. rhizoxinica* or an insect in the case of *Photorhabdus/Xenorhabdus* (*Partida-Martinez et al., 2007*). This raises the intriguing possibility that some of the TRGs in the SOX symbiont genomes might function in protecting the mussel hosts against eukaryotic parasites.

Compared to our knowledge of parasitism in shallow-water bivalves, little is known about parasitism in deep-sea *Bathymodiolus* mussels. This is surprising considering that these incredibly dense communities would be ideal habitats for parasites (*Moreira and López-García, 2003*). Two studies have investigated parasitism in *Bathymodiolus* mussels based on the microscopic identification of unusual 'inclusions' in mussel tissues (*Powell et al., 1999*; *Ward et al., 2004*). The most abundant parasites resembled *Bucephalus*-like trematodes of the phylum Platyhelminthes, which are common in shallow-water mussels (e.g., *Wardle, 1988*; *Lauenstein et al., 1993*; *da Silva et al., 2002*; *Minguez et al., 2012*). *Bucephalus* trematodes infect the gonads of their mussel hosts, which often results in sterilization (*Hopkins, 1957*; *Coustau et al., 1991*). Like their shallow-water relatives, the *Bucephalus*-like trematodes were abundant in the gonads of *Bathymodiolus childressi* from cold seeps in the Gulf of Mexico (*Powell et al., 1999*). *Powell et al. (1999)* estimated that due to this heavy infection, up to 40% of *B. childressi* populations are reproductively compromised.

The distribution of these trematode parasites has not yet been systematically investigated in *Bathymodiolus*. However, of the three species so far studied, only *B. childressi* was infected by trematodes (*Powell et al., 1999*; *Ward et al., 2004*). *B. childressi* is one of the few *Bathymodiolus*

species that only associates with MOX symbionts, but not with SOX symbionts. If many of the TRGs encoded by the *Bathymodiolus* SOX symbiont are being used to defend its host against parasites, as is hypothesized for *Ca.* H. defensa, then this could help to explain why *B. childressi* is so heavily infected by trematodes. The MOX symbionts of *B. azoricus* and *B. childressi* do not encode the abundant TRGs of the *Bathymodiolus* SOX symbiont (Antony CP, personal communication, May 2015).

Our SNP analysis provides further support for the hypothesis that some TRGs may be used for direct beneficial interactions, and some may be used for indirect interactions such as protection against parasites. Genes involved in direct host-symbiont interactions such as recognition and communication are expected to be conserved within the symbiont population of one host species (*Jiggins et al., 2002*; *Bailly et al., 2006*). Consistent with this, eight out of nine genes in the MARTX1 region, which we hypothesize may be involved in attachment to the host, do not have a significantly larger number of SNPs per kb compared to the rest of the genome (*Figure 6—source data 1*). In the ninth gene, annotated as a hypothetical protein, SNPs per kb were slightly above average. In contrast to genes involved in direct host-symbiont interactions, those involved in indirect interactions such as defense against parasites are expected to be highly diverse (see Appendix 2). The large sequence variability in 22 of the TRGs is therefore consistent with a role for these genes in protecting the host against parasites.

## Conclusions

The genomes of the uncultured *Bathymodiolus* SOX symbionts encode a unique arsenal of TRGs, unexpected for a beneficial, nutritional symbiont. We hypothesize that the *Bathymodiolus* SOX symbiont has 'tamed' these genes for use in beneficial interactions with their host. Some of the TRGs may benefit the symbiosis by protecting the symbionts and their hosts from their natural enemies. In most cases, symbionts are either nutritional, that is, their primary role is to provide their host with most or all of its nutrition, or they are defensive (*Douglas, 2014*). The *Bathymodiolus* symbiont is therefore unusual, as it may play an essential role in both nutrition and defense. The TRGs were most likely acquired by HGT, and this may be the mechanism by which its free-living ancestors acquired the ability to form an intimate relationship with marine animals.

Remarkably, the *Bathymodiolus* SOX symbionts encode a larger complement of these TRGs than any so far sequenced pathogen, suggesting that these 'toxins', although initially discovered in pathogens, may in fact belong to larger protein families that function in both beneficial and pathogenic host-microbe interactions. An alternative to the hypothesis that toxins may be tamed for use in beneficial interactions would be that 'symbiosis factors' may be commandeered for use in harmful interactions. Given our recent recognition of the ubiquity and vast natural diversity of mutualistic interactions between bacteria and eukaryotes (*McFall-Ngai et al., 2013*), it is possible that many of the genes that are currently annotated as toxins may have first evolved through beneficial host-microbe associations.

## Materials and methods

### Sampling and processing of *Bathymodiolus* mussels

We collected *Bathymodiolus* mussels in Lilliput, Menez Gwen, and Lucky Strike vent sites on the MAR. *Bathymodiolus* sp. from Lilliput on the SMAR were sampled and processed for genome sequencing as in *Petersen et al. (2011)*. For transcriptomics, we sampled mussels from the SMAR at 09°32.85′S, 13°12.64′W. Specimens of *B. azoricus* were collected in three cruise expeditions to Menez Gwen at (i) 37°45.5777′N, 31°38.2611′W during the MOMARETO cruise, (ii) 37°50.68′N, 31°31.17′W during the RV Meteor cruise M82-3, and (iii) Lucky Strike at 37°16′58.5′′N, 32°16′32.2′′W during the Biobaz Cruise. The adductor muscle was dissected from samples of the MOMARETO cruise, while the gill tissue was dissected from mussels collected during the RV Meteor cruise. For samples from the RV Meteor cruise, we used a combination of differential and rate-zonal centrifugation to enrich *Bathymodiolus* SOX symbiont from gill tissue for genomic and proteomic analyses. Samples for transcriptomics were fixed on board in RNAlater (Sigma, Germany) according to the manufacturer instructions and stored at −80°C. An overview of the samples used in this study is shown in *Supplementary file 1E*.

## DNA extraction, sequencing, genome assembly and binning

DNA extraction and genome sequencing of the *B.* sp. SMAR SOX symbiont was described in *Petersen et al. (2011)*. Briefly, the gill tissue of a single individual was ground in a glass tissue homogenizer and frozen until further processing. In the home laboratory, the homogenate was diluted in phosphate-buffered saline (PBS)1×, centrifuged at 400×*g* for one minute, and the supernatant filtered through a 12-µm GTTP filter (Millipore, Germany). Centrifugation and filtration was repeated 20 times. The filtrates were passed through GTTP filters of 8 µm, 5 µm, 3 µm, and 2 µm. Cells collected on the 0.2-µm filter were used for DNA extraction after *Zhou et al. (1996)* with an initial incubation overnight at 37°C in extraction buffer and proteinase K. 6 kb mate-paired reads were sequenced with 454-Titanium and 36 bp Illumina reads. 454 reads were assembled with Newbler v2.3 (454 Life Sciences Corporation) and 569 pyrosequencing errors were corrected using the Illumina reads.

DNA from the *B. azoricus* SOX symbiont enrichments from three individuals (gradient pellets, see Appendix 4) was extracted according to *Zhou et al. (1996)*. Genomic DNA was extracted from adductor muscle using a CTAB/PVP extraction procedure (2% CTAB, 1% PVP, 1.4 M NaCl, 0.2% beta-mercaptoethanol, 100 mM Tris HCl pH 8, 0.1 mg ml$^{-1}$ proteinase K). After complete digestion of tissues (1 hr at 60°C), the mixture was incubated with 1 µl of RNase for 30 min at 37°C. An equal volume of chloroform-isoamyl alcohol (24:1) was added and tubes were slowly mixed by inversion for 3 min before a 10 min centrifugation at 14,000 rpm and 4°C. The supernatant was collected in a fresh tube, and DNA was precipitated with 2/3 volume of cold isopropanol (1 hr at −20°C). The DNA pellet was recovered by centrifugation (14,000 rpm at 4°C for 20 min), washed with 75% cold ethanol, air-dried, and suspended in 100 µl of sterile water. 454 sequencing was done by Genoscope to sequence the gradient pellet from gill tissue, and by OIST to sequence the adductor muscle of *B. azoricus*. For the gradient pellet, a 3 kb insert 454 library was prepared according to manufacturer protocols for mate-pair sequencing. 630752 reads were generated on a Titanium FLX sequencing machine and assembled using Newbler software (version 08172012). 1310 contigs larger than 500 bp were obtained, forming 130 scaffolds of a total length of 1668565 bp. The assembly from the adductor muscle was done with Newbler v. 2.7 (454 Life Sciences Corporation) as described in *Takeuchi et al. (2012)* resulting in 644000 contigs of a total length of 510449434 bp. The adductor muscle and gradient pellet metagenomes of *B. azoricus* were binned to separate the SOX symbiont from the MOX symbiont and host genomes with Metawatt V. 1.7, which uses tetranucleotide frequencies, coverage, GC content, and taxonomic information for binning (*Strous et al., 2012*). Only sequences longer than 800 bp were considered for further analyses. Since we could only recover an 829 bp fragment of the 16S rRNA from BazSymB, the same DNA that was used for 454 sequencing was used as template for PCR amplification with the universal primers GM3f/GM4r (*Muyzer et al., 1995*). The PCR product was directly sequenced with Sanger and assembled using Geneious V7 (*Kearse et al., 2012*). The 16S rRNA of BazSymB can be found under the accession number (LN871183).

## Genome annotation

We annotated the genomes of the *Bathymodiolus* symbionts (BspSym: PRJNA65421, BazSymA: PRJEB8263, and BazSymB: PRJEB8264), *Candidatus* V. okutanii (NC_009465), *Candidatus* R. magnifica (CP000488), and SUP05 metagenome (ACSG01000000; GQ351266 to GQ351269 and GQ369726) with the following workflow: we used Glimmer (*Delcher et al., 2007*) for open reading frame (ORF) prediction. Ribosomal RNA genes were detected with RNAmmer (*Lagesen et al., 2007*) and tRNAs with tRNAscan-SE (*Lowe and Eddy, 1997*). Annotation was done with GenDB 2.4 (*Meyer, 2003*) and supplemented by JCoast 1.7 (*Richter et al., 2008*) to integrate the results of BLASTp (cut-off e-value of 10.0) against sequence databases NCBI-nr (*Altschul et al., 1997*) SwissProt (*Boeckmann, 2003*), KEGG (*Kanehisa et al., 2011*), COG (*Tatusov, 2000*), Pfam (*Bateman et al., 2004*), and InterPro (*Hunter et al., 2009*). TMHMM (*Krogh et al., 2001*) was used for transmembrane helix analysis and SignalP (*Emanuelsson et al., 2007*) for signal peptide predictions. Sequences of the cytochrome c oxidase subunit 1 of the *Bathymodiolus* mussels were submitted to the European Nucleotide Archive when available (*Supplementary file 1E*).

## PCR amplification of regions with lack of synteny

We designed primers to amplify four regions covering region with lack of synteny. The primer sequences and annealing temperatures are listed in *Supplementary file 1F*. The PCR program

consisted of an initial denaturation step of 98°C for 30 s, followed by 35 cycles at 98°C for 10 s, specific annealing temperature for 30 s, 72°C for 2 min, and a final extension at 72°C for 10 min. We obtained a PCR product of the expected size based on our assembly of all four targeted regions. We sequenced one of these by Sanger sequencing using ABI BigDye v3.1 and the ABI PRISM 3100 genetic analyzer (Applied Biosystems, Foster City, CA).

## Genome analysis and comparison to close relatives

We used CheckM to evaluate the completeness of our draft genomes with a set of single-copy marker genes that are specific to proteobacteria (lineage-specific marker set of CheckM p_Proteobacteria, UID3880) (*Parks et al., 2015*). We estimated the similarity of BazSymB and BazSymA draft genomes with the mean and standard deviation of genes with bi-directional best BLAST hits. The initial comparison of gene content between the clam symbionts, mussel symbionts draft genomes, and SUP05 metagenome was done by BLAST score ratio (BSR) with a BSR cut-off of 0.4 (*Rasko et al., 2005*). Since toxin genes are expected to have a higher mutation rate, we compared the toxin distribution among the three mussel symbiont draft genomes and their closest relatives with a BSR cut-off of 0.2.

Because *Ca.* R. magnifica is the largest vesicomyid genome available, a whole genome comparison with SUP05 was done using a Dotplot produced by Ugene (http://ugene.unipro.ru) with a minimum length of 50 bp and 90% similarity. Since a high gene synteny could be observed, *Ca.* R. magnifica was used as scaffold to reorder the contigs from SUP05 and the *B.* sp. symbiont using Mauve (*Rissman et al., 2009*). We did not do this analysis for the *B. azoricus* symbiont genomes because they were highly fragmented. To compare the influence of HGT on genome evolution, we used the method of *Davis and Olsen (2010)* to identify genes with different codon usage patterns, indicating that they may have been recently acquired via HGT.

## Identification, classification, and structural analysis of TRGs

The initial genome annotation contained many genes annotated as toxins (see Appendix 4 for details of the automatic annotation). Most were related to toxins from three broad previously defined classes, the YD repeats, RTX, and MARTX toxins. Most of the YD repeat and RTX genes could be identified accurately by automatic annotation due to the presence of signature repeat regions. Some of the predicted YD genes appeared to be truncated. Alignment with the closest hits was used to extend the partial ORFs that were not correctly predicted. To curate the annotation of MARTX genes, we built Hidden Markov Model (HMM) profiles for the characterized MARTX domains in *Vibrio cholerae:* actin cross-linking domain, Rho GTPase inactivation, and cysteine protease domain (*Satchell, 2011*). The profiles were used to scan the SOX symbiont genomes with HMMER (*Finn et al., 2011*). We searched for functional domains in the proteins identified as TRGs with SMART (*Ponting et al., 1999*). To analyze the diversity and redundancy of the TRGs in the SOX genomes, we constructed a protein similarity network. All protein sequences of the SOX symbionts were searched against Uniprot with BLAST (coverage >50%, similarity >25%, e-value < e−5). We also used BLAST of all against all sequences of the *Bathymodiolus* SOX symbiont to recover partial TRGs that could not be identified by automatic annotation. The two searches were combined to produce a sequence similarity network based on transitivity clustering using the plugin Blast2SimilarityGraph in Cytoscape (*Srinivasan and Moon, 1999*; *Shannon et al., 2003*; *Wittkop et al., 2010*). Only the sub-networks that were connected in a maximum of two steps to the TRGs were considered. All TRGs of the SOX symbionts were submitted to Phyre2 (*Kelley and Sternberg, 2009*) to predict the secondary structure of the protein. We looked for clusters in the protein similarity network that could be associated with a subunit of a toxin complex or an active domain.

## Search for TRGs in SUP05

None of the three toxin-related classes has characteristic patterns, profiles, or domains that can be searched with standard tools. Often, their only shared feature is a repeat region, such as the RTX repeat in RTX and MARTX, which is a calcium-binding site containing G and N residues. However, the number of G and N residues and the length of the repeat is highly variable, which makes profile searches impossible. Moreover, these repeat regions can be shared by other calcium-binding proteins such as integrins and fibronectin, which do not act as toxins. The sequence and length of the YD

repeat is similarly variable. To identify homologs of the *Bathymodiolus* symbiont TRGs in published metagenomes and metatranscriptomes, we therefore used BLASTp (coverage >50%, similarity >25%, e-value < 0.001) (*Supplementary file 1G*).

Best hits were blasted against the SOX symbiont proteins to search for signatures of the TRGs. We looked for bacterial genomes that had similar TRGs to the *Bathymodiolus* SOX symbiont, using the Integrated Microbial Genomes (IMG) database (*Markowitz et al., 2011*) with BLASTp (>50% coverage, > 25% similarity, e-value 0.001). 122 genomes were retrieved and manually curated for the potential of pathogenicity and biofilm formation, as well as the lifestyle categories extracellular host-associated, intracellular, and free-living bacteria. Kruskall–Wallis and Mann–Whitney–Wilcoxon tests were used to estimate if the increased number of a class of TR is biased in a certain lifestyle. All statistical analyses were done in R.

## Phylogenetic analyses

The 16S rRNA gene sequences of the *Bathymodiolus* symbionts were imported into the Silva database (release Ref 119) and initially aligned with the SINA aligner (*Pruesse et al., 2012*). The final alignment was refined with MAFFT (*Katoh et al., 2002*). A maximum likelihood tree was estimated from the alignment of 1653 nucleotide positions using RaxML with 100 bootstrap replicates.

To construct the YD phylogeny, the data set from BspSym was used to obtain related sequences from GenBank and *B. azoricus*. Because two metagenomes of *B. azoricus* were analyzed, we used CD-hit to remove redundancy at 100% similarity (*Li and Godzik, 2006*). YD repeat proteins are very variable in the C and N terminus. Therefore, the selection criterion we used was the presence of the conserved *rhs* domain. Only this 'core' region was used for phylogeny (*Jackson et al., 2009*). YD proteins were aligned with MAFFT with BLOSUM30 (*Katoh et al., 2002*). Phylogenetic analyses were done in ARB with maximum likelihood and bayesian reconstructions using a filter of 10% similarity, which resulted in 536 amino acid positions (*Ronquist and Huelsenbeck, 2003*; *Ludwig et al., 2004*; *Stamatakis, 2006*). Bootstrap support was calculated with 100 replicates in maximum likelihood. MrBayes was run for 9 million generations and two independent runs of four heated chains. A consensus tree of both methods was constructed. Polytomies were introduced when both methods did not agree.

## Statistical analyses of TRGs content in microbial genomes

Genomes with curated metadata are available through IMG (*Markowitz et al., 2011*). Nevertheless, most entries contained no information in categories we were interested in such as biofilm formation and intracellular lifestyle. IMG was used as starting point to search for curated genomes that had genes with similarity to the TRGs of the mussel symbionts (>10% similarity, e-value < $e^{-5}$). The protein sequences of these genomes were retrieved from NCBI based on the genome name (modified Python script from Sixing Huang, Max Planck Institute Bremen). BLASTp was used to retrieve protein sequences related to the SOX TRGs (>50% coverage, > 25% similarity, e-value of 0.001). 122 genomes had at least one gene that was similar to at least one TRG. We considered the genomes of different strains as independent events for statistical analysis, as even closely related strains can have different lifestyles. These genomes were manually curated based on a literature search and classified according to the following categories: (1) lifestyle: divided into three sub-categories (a) associated: if the organism is at any stage host-associated (based on IMG metadata), (b) intracellular: includes obligate and facultative symbionts (searched in Google scholar with the keywords 'intracellular bacteria' and references read for more details when the abstract was not sufficient), (c) free-living: bacteria that are not host-associated and not intracellular; (2) pathogen: bacteria that can produce disease (information obtained from IMG); (3) biofilm: bacteria found in biofilms (Google scholar keywords 'biofilm' and 'microbial mat' with literature analyses when not clear).

The data were formatted and merged with self-written Perl scripts and *R Development Core Team (2011)*. Genomes with similar toxins or TRGs to the *Bathymodiolus* symbionts are shown in *Supplementary file 1B*. The sum of the number of TRGs belonging to YD, RTX, and MARTX was normalized with the total gene count for each genome and multiplied by a factor of 1000. To compare the number of TRGs against the lifestyle categories, we used Kruskall–Wallis test. A post hoc analysis was carried out on significant p-values for the associated category with the Mann–Whitney–Wilcoxon test. Statistical analyses were done in R. We tested whether the bacteria were enriched in any of the

three toxin-related classes at the class, order, or family by using one-way Permanova with 9999 permutations and Euclidean distances. p-values were corrected with Bonferroni correction for multiple testing.

## Transcriptomics

To extract the total RNA of three individuals of *B. azoricus*, the gill tissue was incubated overnight in RNAlater (Sigma) at 4°C. A fragment of the gill was dissected and homogenized. RNA was extracted with RNeasy Plus MicroKit (Qiagen, Hilden, Germany) according to the manufacturer's instructions. To remove cell debris and to improve RNA yield, we used QIAshreder Mini Spin Columns (Qiagen). The quality of the RNA was assessed with Agilent 2100 Bioanalyzer. The RNA was used for cDNA synthesis with the Ovation RNA-Seq System V2 (NuGEN, San Carlos, CA). To extract the total RNA of three individuals of *B.* sp., the gill tissue was placed separately on liquid nitrogen, homogenized, and stored overnight at 4°C in self-made RNAlater (10 mM ethylenediaminetetraacetic acid (EDTA), 25 mM tri-sodiumcitrate-dihydrate, 5.3 M ammonium sulfate, adjusted to pH 5.2). After removal of RNAlater, samples were incubated in 600 µl RLT-β-mercaptoethanol buffer (1:100) for 10 min and homogenized on QIAshredder columns (Qiagen). Total RNA was extracted with RNeasy Mini Kit (Qiagen). We applied the DNA-free DNAse Treatment and Removal Kit according to the manufacturer's instructions (Invitrogen, Carlsbad, CA/Ambion, Austin, TX). RNA quality was checked on the Experion Automated Electrophoresis Station using the RNA StdSens Analysis protocol (BioRad, Hercules, CA).

Libraries of *B.* spp. were generated with the Illumina TruSeq RNA Sample Preparation Kit and sequenced 2 × 100 paired-end on an Illumina HiSeq 2000 platform at the Institute of Clinical Molecular Biology (Kiel). A total of 32.9, 38.2, and 38.4 million reads were sequenced per individual of *B. azoricus*. Libraries of *B. azoricus* were generated with DNA library prep kit for Illumina (BioLABS, Frankfurt am Main, Germany) and sequenced as single 100-bp reads on an Illumina HiSeq 2500 platform at the Max Planck Genome Centre (Cologne). A total of 4.3, 4.8, and 6.9 million reads were sequenced per individual of *B. azoricus*. Adaptor removal and quality trimming was done with Nesoni (http://www.vicbioinformatics.com/software.nesoni.shtml) using a quality threshold of 20. To remove ribosomal sequences from the data, we mapped the reads against the SILVA 115 SSU database with Bowtie2 and kept those reads that failed to align (*Langmead and Salzberg, 2012*; *Quast et al., 2013*). The abundance of the transcripts per gene was estimated with Rockhopper that uses upper quartile normalization (*McClure et al., 2013*). The expression values of the TRGs were normalized to the expression of RubisCO (*Supplementary file 1D*). Transcriptome reads that mapped to the SOX symbionts with Bbmap (http://bbmap.sourceforge.net/) were deposited in the European Nucleotide Archive under the accession numbers PRJEB7941 for *B. azoricus* and PRJEB7943 for *B.* sp.

## SNP analysis

We calculated SNPs to compare substitution rates of genes in the SOX symbiont genomes using three transcriptomes of *B.* sp and three of *B. azoricus* (*Supplementary file 1E*). Reads were normalized to a coverage of maximum 200 and minimum of five with BBNorm v33 (http://sourceforge.net/projects/bbmap/). Transcriptomes of *Bathymodiolus* sp. were mapped against the draft genome BspSym and transcriptomes of *B. azoricus* were mapped to the draft genomes of BazSymA and BazSymB with BBMap v33. We only considered those reads that mapped to a single position in the reference genome and that had higher than 90% identity alignments. SNPs were called independently for the draft genomes BspSym, BazSymA, and BazSymB with the Genome Analysis ToolKit as described by *De Wit et al. (2012)* with some modifications (*McKenna et al., 2010*). In summary, regions needing realignment were identified and realigned over intervals. SNPs and insertions or deletions (InDels) were called with the haplotype caller with a minimum confidence of 20. A filter was applied around InDels with a mask extension of 5. SNPs per gene were obtained with BEDTools (*Quinlan and Hall, 2010*). The number of SNPs per gene was normalized according to the gene length minus regions of unknown sequence for genes containing Ns. We did not consider genes shorter than 150 bp nucleotides or outlier genes at the ends of scaffolds that had an unusual number of SNPs.

## Proteomics

Soluble proteins were extracted from SOX symbiont enrichments, host cytosolic fractions, and whole gill and foot tissue in biological duplicates (Appendix 4). All proteome samples were obtained from the M82-3 cruise. Membrane proteins were extracted from the SOX symbiont enrichments and the

whole gill tissue samples. We used 1D-PAGE followed by liquid chromatography (1D-PAGE-LC) to separate proteins and peptides as described previously with minor modifications (*Heinz et al., 2012*). MS/MS spectra were acquired with a LTQ Orbitrap Velos mass spectrometer (Thermo Fisher, Bremen, Germany) for soluble proteins and a LTQ Orbitrap Classic (Thermo Fisher Scientific Inc., Waltham, MA) for membrane proteins (Appendix 4). MS/MS data were searched against two databases using the SEQUEST algorithm (*Eng et al., 1994*). The first database, designated 'reduced' database, contained protein sequences from the SOX and MOX symbionts from *Bathymodiolus*, as well as from the host (see Appendix 4). The second database contained in addition sequences from host-related bivalves and symbiont-related bacteria (*Supplementary file 1H*). False discovery rates were determined with the target-decoy search strategy as described by Elias and Gygi (*Elias and Gygi, 2007*; *Kleiner et al., 2012a*).

## Acknowledgements

We thank the captains and crews of the research vessels and ROVs involved in the sampling effort, including Francois Lallier and Arnaud Tanguy who provided samples for sequencing from the MoMARETO cruise (*Sarrazin et al., 2006*) with the N/O *Pourquoi Pas?* and ROV *Victor 6000*. This project was funded by the Max Planck Society, an ERC Advanced Grant and a Gordon and Betty Moore Foundation Marine Microbial Initiative Investigator Award to ND, the DFG Cluster of Excellence 'The Ocean in the Earth System' at MARUM, Bremen, a DAAD scholarship to LS, OIST funding to NS, and the Marie Curie Initial Training Network 'Symbiomics' (project no. 264774) for the proteomics. We thank Harald Gruber-Vodicka, Hanno Teeling, and Brandon Seah for useful discussions, Michael Richter for his help with GenDB, and Sixing Huang for providing a Python script. We thank the three reviewers whose excellent suggestions helped us to improve the manuscript.

## Additional information

### Funding

| Funder | Grant reference | Author |
|---|---|---|
| Max-Planck-Gesellschaft (MPG) | | Lizbeth Sayavedra, Manuel Kleiner, Silke Wetzel, Dennis Fink, Nicole Dubilier, Jillian M Petersen |
| Gordon and Betty Moore Foundation | MMI Investigator Grant 3811 | Nicole Dubilier |
| European Research Council (ERC) | Bathybiome | Lizbeth Sayavedra, Manuel Kleiner, Silke Wetzel, Dennis Fink, Nicole Dubilier, Jillian M Petersen |
| Deutsche Forschungsgemeinschaft (DFG) | | Lizbeth Sayavedra, Manuel Kleiner, Silke Wetzel, Nicole Dubilier, Jillian M Petersen |
| German Academic Exchange Service (DAAD) | | Lizbeth Sayavedra |
| European Research Council (ERC) | Symbiomics ITN 264774 | Ruby Ponnudurai, Thomas Schweder, Stephanie Markert |
| Okinawa Institute of Science and Technology Graduate University (OIST) | | Nori Satoh |

The funders had no role in study design, data collection and interpretation, or the decision to submit the work for publication.

### Author contributions

LS, Wrote the paper; analyzed genomic and transcriptomic data, Conception and design, Acquisition of data, Analysis and interpretation of data, Drafting or revising the article; MK, RP, Performed proteomic analyses and processed mass spectrometry data, Drafting or revising the article; SW, Developed symbiont extraction method, Drafting or revising the article; EP, VB, Sequenced and assembled samples sent to Genoscope, Drafting or revising the article; NS, ES, Sequenced and

assembled samples in OIST, Drafting or revising the article; DF, DF performed gradient centrifugation during the M82-3 cruise, Drafting or revising the article; CB, TBHR, PR, MBS, Sequenced transcriptomes from B. sp., Drafting or revising the article; DB, TS, SM, Performed proteomic analyses and processed mass spectrometry data, Drafting or revising the article; ND, Conception and design, Drafting or revising the article; JMP, Wrote the paper, Conception and design, Drafting or revising the article

## Additional files

### Supplementary file

• Supplementary file 1. (A) Number of mobile elements in the genomes compared in this study. (B) Genomes with toxin-related genes (TRGs) similar to those of the sulfur-oxidizing (SOX) symbionts of *Bathymodiolus*. The number of genes per TRGs class is shown. (C) p-values obtained with one-way Permanova were corrected with Bonferroni correction for multiple testing. Number of TRGs per genome was normalized to the total gene count. (D) Transcriptome counts of three individuals from *B. azoricus* and three individuals from *B*. sp. were mapped to their respective reference genomes with Rockhopper. Expression values of TRGs were normalized to the expression of RubisCO. (E) Samples used in this study. (F) Primer sequences and annealing temperatures used to detect genome rearrangements. (G) Metagenomes and metatranscriptomes enriched in SUP05 from oxygen minimum zones (OMZ) or hydrothermal vents. (H) Amino acid sequences from the following genomes were used in the reference database for proteomic analysis (lncDB). The genomes belong to relatives of the SOX and methane-oxidizing (MOX) symbionts of *B. azoricus*, as well as the mussel host. (I) Details of expressed toxin-related proteins identified with proteomics. The values are given in % normalized spectral abundance factor (NSAF), which is a normalized spectral abundance factor that gives the relative abundance of a protein in a sample in %.

### Major datasets

The following datasets were generated:

| Author(s) | Year | Dataset title | Dataset ID and/or URL | Database, license, and accessibility information |
|---|---|---|---|---|
| Sayavedra L, Kleiner M, Ponnudurai R, Wetzel S, Pelletier E, Barbe V, Satoh N, Shoguchi E, Fink D, Breusing C, Reusch TBH, Rosenstiel P, Schilhabel M, Becher D, Schweder T, Markert S, Dubilier N, Petersen JM | 2015 | Endosymbiont of Bathymodiolus sp. | http://www.ebi.ac.uk/ena/data/view/PRJNA65421 | Publicly available at the EBI European Nucleotide Archive (Accession no: PRJNA65421). |
| Sayavedra L, Kleiner M, Ponnudurai R, Wetzel S, Pelletier E, Barbe V, Satoh N, Shoguchi E, Fink D, Breusing C, Reusch TBH, Rosenstiel P, Schilhabel M, Becher D, Schweder T, Markert S, Dubilier N, Petersen JM | 2015 | Genome of Sulfur-oxidizer endosymbiont of Bathymodiolus azoricus from Menez Gwen (BazSymA) | http://www.ebi.ac.uk/ena/data/view/PRJEB8263 | Publicly available at the EBI European Nucleotide Archive (Accession no: PRJEB8263). |
| Sayavedra L, Kleiner M, Ponnudurai R, Wetzel S, Pelletier E, Barbe V, Satoh N, Shoguchi E, Fink D, Breusing C, Reusch TBH, Rosenstiel P, Schilhabel M, Becher D, Schweder T, Markert S, Dubilier N, Petersen JM | 2015 | Genome of Sulfur-oxidizer endosymbiont of Bathymodiolus azoricus from Menez Gwen (BazSymB) | http://www.ebi.ac.uk/ena/data/view/PRJEB8264 | Publicly available at the EBI European Nucleotide Archive (Accession no: PRJEB8264). |

| Author(s) | Year | Dataset title | Dataset ID and/or URL | Database, license, and accessibility information |
| --- | --- | --- | --- | --- |
| Sayavedra L, Kleiner M, Ponnudurai R, Wetzel S, Pelletier E, Barbe V, Satoh N, Shoguchi E, Fink D, Breusing C, Reusch TBH, Rosenstiel P, Schilhabel M, Becher D, Schweder T, Markert S, Dubilier N, Petersen JM | 2015 | Mitochondrial COI gene for cytochrome oxidase subunit 1 | http://www.ebi.ac.uk/ena/data/view/LN833433-LN833440 | Publicly available at the EBI European Nucleotide Archive (Accession no: LN833433-LN833440). |
| Sayavedra L, Kleiner M, Ponnudurai R, Wetzel S, Pelletier E, Barbe V, Satoh N, Shoguchi E, Fink D, Breusing C, Reusch TBH, Rosenstiel P, Schilhabel M, Becher D, Schweder T, Markert S, Dubilier N, Petersen JM | 2015 | Transcriptome of Sulfur-oxidizer endosymbiont of Bathymodiolus azoricus | http://www.ebi.ac.uk/ena/data/view/PRJEB7941 | Publicly available at the EBI European Nucleotide Archive (Accession no: PRJEB7941). |
| Sayavedra L, Kleiner M, Ponnudurai R, Wetzel S, Pelletier E, Barbe V, Satoh N, Shoguchi E, Fink D, Breusing C, Reusch TBH, Rosenstiel P, Schilhabel M, Becher D, Schweder T, Markert S, Dubilier N, Petersen JM | 2015 | Transcriptome of Sulfur-oxidizer endosymbiont of Bathymodiolus sp. 9 South | http://www.ebi.ac.uk/ena/data/view/PRJEB7943 | Publicly available at the EBI European Nucleotide Archive (Accession no: PRJEB7943). |
| Sayavedra L, Kleiner M, Ponnudurai R, Wetzel S, Pelletier E, Barbe V, Satoh N, Shoguchi E, Fink D, Breusing C, Reusch TBH, Rosenstiel P, Schilhabel M, Becher D, Schweder T, Markert S, Dubilier N, Petersen JM | 2015 | Bathymodiolus azoricus thioautotrophic gill symbiont partial 16S rRNA gene, isolate BazSymB | http://www.ebi.ac.uk/ena/data/view/LN871183 | Publicly available at the EBI European Nucleotide Archive (Accession no: LN871183). |

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

## Appendix 1

### The role of the symbionts in the *Bathymodiolus* association

In their complement of toxin-related genes (TRGs), the *Bathymodiolus* symbiont genomes closely resemble those of pathogenic bacteria. In fact, it has a larger arsenal of these particular types of TRGs than any so far sequenced pathogen. Considering this, is it possible that their role in the association with their mussel hosts has been misunderstood? Could they in fact be pathogens of *Bathymodiolus* rather than beneficial symbionts? Chemosynthetic symbionts are widely assumed to benefit their hosts by contributing a source of organic matter for their nutrition and so far this was also thought to be true for the *Bathymodiolus* symbioses (*Van Dover, 2000*; *Cavanaugh et al., 2006*; *Dubilier et al., 2008*). There is a variety of evidence supporting that the *Bathymodiolus* symbionts are in fact an effective source of nutrition, including: (1) *Bathymodiolus* mussels can grow to densities two to five times higher than their shallow-water relatives that rely exclusively on filter-feeding for their nutrition (*Gebruk et al., 2000*), (2) *Bathymodiolus* mussels have a reduced feeding groove and digestive system compared to their filter-feeding relatives (Gustafson et al., 1998), (3) carbon transfer from sulfur-oxidizing (SOX) symbionts to *B. azoricus* has been demonstrated experimentally (*Riou et al., 2010*), and (4) based on their genomes, the symbionts are autotrophs (see below). These observations are all consistent with the hypothesis that the SOX symbionts benefit their *Bathymodiolus* hosts by providing them with a source of nutrition.

### Symbiont metabolism

The symbionts encode a modified version of the Calvin cycle that lacks the enzyme sedoheptulose-1,7-biphosphatase and fructose-1,6-bisphosphatase. Instead, the symbionts have a pyrophosphate-dependent phosphofructokinase (PPi-PFK) as described in other chemosynthetic symbioses (*Markert et al., 2011*; *Kleiner et al., 2012b*; *Dmytrenko et al., 2014*). In contrast to the clam symbionts and SUP05 that possess ribulose bisphosphate carboxylase/oxidase (RubisCO) form II, *Bathymodiolus* symbionts have RubisCO form I. RubiscCO form I is better adapted to higher $O_2$ concentrations and has a medium to low affinity to $CO_2$, indicating an adaptation to environments with $O_2$ present (*Cavanaugh and Robinson, 1996*; *Badger and Bek, 2008*).

The genome did not provide sufficient evidence to determine whether the *Bathymodiolus* symbionts are obligate autotrophs or mixotrophs. None of the three essentially complete symbiont genomes encode the enzyme alpha-ketoglutarate dehydrogenase, which is often absent in the genomes of obligate autotrophs (*Wood et al., 2004*). Some other genes of the tricarboxylic acid (TCA) cycle were also missing, but this may be because the genomes are not yet closed. A few tripartite ATP-independent periplasmic (TRAP) transporters were present, and these could be involved in uptake of organic compounds.

To acquire the necessary energy to fuel carbon fixation, the *Bathymodiolus* symbionts can utilize reduced sulfur compounds and hydrogen (*Petersen et al., 2011*; *Kleiner et al., 2012a*). The pathways for sulfur oxidation present in the genomes are similar to those of other SOX symbionts of *R. pachyptila* tubeworms, or vesicomyid and solemyid clam symbionts (*Figure 2—figure supplement 3*) (*Robidart et al., 2008*; *Harada et al., 2009*; *Kleiner et al., 2012a*; *Dmytrenko et al., 2014*). We found the genes that encode SoxABXYZ and sulfide-quinone reductases (Sqr), which reduce cytochrome c and quinones using sulfide and/or thiosulfate as electron donor. The *Bathymodiolus* SOX symbionts might store elemental sulfur based on the absence of SoxCD sulfur dehydrogenase, although the symbionts lack the large intracellular inclusions typical of sulfur storage, and elemental sulfur has not yet been detected in mussel gills. Sulfide can be oxidized to sulfite with the reverse dissimilatory sulfite reduction pathway (rDsr), and sulfite can be further oxidized to sulfate with APS reductase and an ATP-generating ATP sulfurylase (Sat) (Dahl et al., 2008).

Besides the use of oxygen as electron acceptor, we found in the most complete genome of the symbiont from *B. azoricus* the genes for the use of nitrate as electron acceptor (nitrate reductase—*narGHIJ*); however, we did not find these genes in the other two symbiont genomes. The genes required to convert nitrite to nitrous oxide were found in all three-draft genomes (nitrite reductase—*nirK* and nitrite oxidoreductase—*norCB*). We might have not found the nitrate reductase in the draft genomes of BspSym and BazSymB due to the incompleteness (**Table 1**). The use of an alternative electron acceptor could reduce the competition for oxygen with their host.

## Appendix 2

# Other potential roles of the TRGs

### RTX toxins as nutrient-scavenging proteins

RTX genes were not preferentially found in host-associated bacteria but were also regularly found in free-living bacteria. Bloom-forming aquatic cyanobacteria had some of the highest numbers of RTX genes (*Figure 5*, *Figure 5—figure supplement 3*). Cyanobacteria are well-known producers of secondary metabolite toxins called cyanotoxins, but protein toxins of cyanobacteria have received far less attention (*Kaebernick et al., 2000*). A hemolysin-like protein from *Synechocystis* sp. PCC 6803 was shown to be located in the S-layer of *Synechocystis* cells, and its deletion causes cells to be more permeable and more sensitive to environmental toxins (*Liu et al., 2011*; *Sakiyama et al., 2011*). This hemolysin-like protein therefore seems to have a protective rather than a toxic function, although it did have slight hemolytic activity when tested against sheep erythrocytes (*Sakiyama et al., 2006*). However, the function of most cyanobacterial RTX toxins has not yet been investigated. Intriguingly, cyanobacterial blooms are tightly coupled to nutrient availability (*Paerl, 2008*). If free-living cyanobacteria can use RTX proteins as toxins to obtain nutrients by lysing other organisms, analogous to pathogens lysing host cells, they could gain access to additional sources of nutrients.

Members of the marine *Roseobacter* clade also had an unusually large number of RTX genes (*Figure 5—figure supplement 3*). *Roseobacter* are some of the most abundant bacteria in marine habitats. They commonly associate with a range of eukaryotic organisms from single-celled phytoplankton to vertebrate and invertebrate animals (*Buchan et al., 2005*; *Wagner-Döbler and Biebl, 2006*; *Brinkhoff et al., 2008*). Previous studies have shown that the RTX genes of *Roseobacter* are expressed, and that they can make up to 90% of the excreted proteome (*Christie-Oleza et al., 2012*). However, their function in *Roseobacter* is still unknown.

### Could strain variation of *Bathymodiolus* symbionts be promoted by a defensive role of TRGs?

A role for the *Bathymodiolus* SOX symbiont in host defense against pathogens could help to explain why the mussels seem to associate with multiple closely related SOX strains (*Duperron et al., 2006*). Associations with multiple very similar symbiont strains are considered evolutionarily unstable because they promote the emergence of 'cheaters', symbionts that benefit from the association but do not contribute to the cost (*West et al., 2002*; *Hart et al., 2013*). For example, a 'cheater' could be a SOX symbiont that occupies space within the host, thus gaining access to the electron donors and acceptors it needs for its growth, but passes on less organic matter to the host than neighboring 'cooperators'. Cheaters have a distinct growth advantage, and could easily outcompete cooperators, ultimately leading to the breakdown of the association. Despite this, multiple strains of SOX bacteria can occur in individual *Bathymodiolus* mussels (*Won et al., 2003a*; *DeChaine et al., 2006*). Intriguingly, the SOX symbiont strain variation identified in genetic markers such as the 16S-ITS-23S rRNA gene operon is similar to the strain variation seen in natural populations of defensive *Ca.* H. defensa symbionts (*Russell et al., 2013*). Moreover, we have shown that closely related SOX symbionts show extensive sequence variation in most TRGs, which is also the case for *Ca.* H. defensa (*Russell et al., 2013*). This large natural diversity is inconsistent with a role for the TRGs in host-symbiont recognition and communication, as genes mediating these processes should be conserved (*Jiggins et al., 2002*; *Bailly et al., 2006*). However, this diversity could be explained by antagonistic co-evolution between protective symbionts and their host's parasites. Indeed, experimental evolution shows that natural enemies quickly acquire resistance to *Ca.* H. defensa (*Rouchet and Vorburger, 2014*). However, this resistance is highly specific to the strain of *Ca.* H. defensa, the parasite is

experimentally challenged with and is useless against other closely related strains (***Schmid et al., 2012***; ***Rouchet and Vorburger, 2014***). Associating with diverse symbiont strains, each with its own unique arsenal of TRGs, could therefore protect the host in the face of rapidly evolving parasite resistance.

## Appendix 3

### Structure of *Bathymodiolus* symbiont TRGs and genomic regions

In BspSym, the 19 multifunctional autoprocessing RTX (MARTX)-like genes were clustered in two genomic regions that spanned 58.5 kb (MARTX1) and 46.3 kb (MARTX2). Genes predicted in these regions encoded many of the unique features of MARTX genes and other large repetitive RTX proteins including RTX repeats, cysteine protease domains (CPDs), hemagglutinin, and adhesion domains (*Figure 4—figure supplement 2*). Moreover, some of these genes were very large, which is typical of MARTX and MARTX-like genes (*Satchell, 2007*, *2011*). BazSymA and BazSymB also encoded genes with characteristic signatures of MARTX genes, but these two draft genomes were highly fragmented, and the MARTX genes were found on very small genome fragments.

Both of the BspSym MARTX regions contained genes encoding RTX repeats and RTX CPDs. The RTX CPD has so far only been found in MARTX genes. Its function is to cleave off effector domains, whose sequence and effects on eukaryotic hosts can vary substantially even in MARTX genes from different strains of the same bacterial species (*Roig et al., 2010*; *Kwak et al., 2011*). Beyond these shared features, MARTX1 and MARTX2 of the BspSym had distinct differences. Genes in MARTX1 encoded six hemagglutinin domains, two adhesion domains, and a pectin lyase domain. Similar features are found in large repetitive RTX adhesins, except that the CPD domain is not usually found in this protein subfamily (*Satchell, 2011*). In fact, the combination of hemagglutinin, pectin lyase, and CPD domains has so far only been found in the filamentous hemagglutinin protein of the betaproteobacterial pathogens *B. pertussis* and *B. bronchiseptica*, which is a key determinant of epithelial cell attachment and immune response modulation in *Bordetella* infections (*Melvin et al., 2014*). In contrast to *Bordetella*, where these domains are all encoded in one very large ORF, these domains were encoded in neighboring ORFs in the *Bathymodiolus* symbiont. However, MARTX-encoding genes are notoriously difficult to predict due to their repetitive structure (*Satchell, 2007*, *2011*). We therefore cannot exclude the possibility that these may in fact encode one large protein in the *Bathymodiolus* symbiont.

The second MARTX-like gene cluster in the BspSym genome, MARTX2, resembles a 'typical' hemolysin operon. It contains all the necessary genes for type I secretion, and a gene annotated as *rtxA*, which encodes a hemolysin toxin. However, in contrast to typical hemolysin operons, it is missing the hemolysin activator gene. Interestingly, MARTX1, but not MARTX2, contains a gene annotated as a hemolysin activator. Moreover, the MARTX2 region contains four genes that each has up to five CPD domains. The CPD domain is typical for MARTX genes but not for known hemolysins. This region therefore has features characteristic of both RTX and MARTX genes.

## Appendix 4

# Supplementary materials and methods

### Symbiont and host separation via density gradient centrifugation for genomics and proteomics from gill tissue

To separate the SOX symbionts from gill tissues and to obtain a fraction enriched in host cytosolic proteins from the gill, a combination of differential and rate-zonal centrifugation was applied. All steps were carried out at 4°C. Around 1 g of gill tissue was removed from a mussel and transferred into a Duall homogenizer (Tissue grind pestle and tube SZ22, Kontes Glass Company, Vineland, New Jersey). The tissue was covered with 1× PBS and homogenized thoroughly with about 20 strokes. To remove host nuclei and large tissue fragments, the homogenate was filled into a 15 ml conical tube, 1× PBS added up to 15 ml, and centrifuged for 10 min at 700×$g$ in a swing out rotor. The supernatant was carefully transferred into a new 15 ml tube and the pellet (first pellet) frozen at −80°C. The supernatant was centrifuged again for 10 min at 700×$g$ in a swing out rotor to remove any remaining nuclei and tissue fragments. The resulting supernatant was transferred to a new tube and centrifuged in a fixed angle rotor for 10 min at 15.000×$g$ to pellet all symbionts, mitochondria, and small tissue fragments. The supernatant from this centrifugation step, which contained the cytosolic host proteins, was frozen at −80°C. The pellet was resuspended in 1× PBS and the suspension was layered on top of a discontinuous density gradient made with HistoDenz (Sigma) dissolved in 1× PBS. The density gradient was set up in 5% steps from 5 to 25% (wt/vol) HistoDenz. The density gradient was centrifuged in a swing out rotor for 7 min at 3000×$g$ and was then divided into equally sized fractions. The pellet in the gradient tube (gradient pellet) was frozen at −80°C until processing. The gradient fractions were washed twice with 1× PBS to remove the HistoDenz and the resulting pellets were frozen at −80°C. Subsamples for catalyzed reporter deposition-fluorescence in situ hybridization (CARD-FISH) analyses were taken from the homogenates, pellets, and gradient fractions prior to freezing for later analysis of sample composition.

### Analysis of symbiont composition in density gradient fractions using CARD-FISH

To determine the abundance and composition of host material and the different symbiont species in the homogenates, supernatants, pellets, and gradient fractions, we used CARD-FISH. Subsamples of gradient pellets were fixed overnight in 1% formaldehyde at 4°C, washed three times in 1× PBS, and stored at −20°C in 50% 1× PBS and 50% ethanol. In the home laboratory, cells were poured through 0.22-µm filters. CARD-FISH was done as described previously (*Pernthaler et al., 2002*) with minor modifications. Endogenous peroxidases were inactivated with 0.01 M HCl for 10 min at room temperature. Double hybridization of rRNA was performed with the probes BMARt-193 for the SOX symbiont and BMARm-845 for the methane-oxidizing (MOX) symbionts (*Duperron et al., 2006*). Sections were hybridized at 46°C for 2 hr, followed by washing at 48°C for 20 min. Amplification was done for 10 min at 37°C, followed by inactivation of horseradish peroxidase (HRP) using methanol. Prior to microscopic evaluation, the cells were counterstained with 1 µg ml⁻¹ 4′,6-diamidino-2-phenylindole (DAPI). Cells with a symbiont-specific signal were counted against at least 500 DAPI signals per filter section.

### Proteomic analysis

#### Extraction of cytosolic and membrane proteins

For the proteomic analyses, gradient pellets, host supernatant (described above), and frozen tissues from mussel gill and foot were processed in biological duplicates. For washing, the samples were resuspended in TE buffer (10 mM Tris pH 7.5, 10 mM EDTA pH 8.0, 1× complete Protease Inhibitor Cocktail [Roche Applied Science, Germany]) and centrifuged briefly (3 min, 4°C and 21,500×$g$). The resultant pellets were homogenized using a Duall homogenizer and the

homogenate was transferred to low binding 1.7 ml reaction tubes (Sorenson BioScience Inc., Salt Lake City, UT). Cells were lysed on ice using a sonicator (Bandelin Sonopuls ultrasonic homogenizer, Germany); 2 × 25 s at 30% power and a cycle of 0.5 s, with a 30 s pause. The lysate was centrifuged to remove cell debris (10 min, 4°C and 15,300×*g*). The resulting supernatants, that is, protein raw extracts, were further subjected to ultracentrifugation (100,000×*g* for 60 min at 4°C), allowing for the enrichment of membranes and membrane-associated proteins in the pellets and accumulation of soluble proteins in the supernatants. Protein concentrations in the soluble protein extracts were determined using the method described by *Bradford, 1976* and aliquots for all four sample types (symbiont-enriched gradient pellet, host supernatant, gill tissue, and foot tissue) were stored at −80°C until MS analysis. Additionally, for gill tissue and gradient pellet samples, membrane proteins were purified from the enriched membrane fraction (see above) according to the protocol of *Eymann et al. (2004)* as described by *Markert et al. (2007)*. The purified membrane protein fraction (in 30 μl of 50 mM triethylammonium bicarbonate buffer, pH 7.8) was transferred to low binding 1.7 ml tubes and immediately used for MS analysis. Due to sample scarcity, the biological duplicates for these membrane protein samples were pooled together to obtain enough protein for MS analysis.

## 1D-PAGE-LC-MS/MS

Approximately, 20 μg of protein of the membrane and soluble protein extracts, respectively, was dissolved in 20 μl of sample loading buffer (100 mM Tris-HCl (pH 6.8), 10% sodium dodecyl sulfate (SDS), 20% glycerol, 5% β-mercaptoethanol, 0.1% bromophenol blue). Proteins were separated by one-dimensional (1D) SDS PAGE in pre-cast 10% polyacrylamide gels (BioRad), fixed and stained with Coomassie Brilliant Blue (G250, Sigma-Aldrich, Germany). The gel lanes were cut into 10 equal sized pieces, destained with washing buffer (200 mM ammonium hydrogen carbonate, 30% acetonitrile), dried and proteins were subjected to overnight in-gel digestion with trypsin (Promega, Germany) (*Heinz et al., 2012*). Prior to mass spectrometric analysis, the peptides were purified using Ziptips (P10, U-C$_{18}$, Millipore).

Tryptic digests of the cytosolic samples were subjected to liquid chromatography performed on an EASYnLC (Proxeon, Denmark) with self-packed columns (Luna 3μ C18(2) 100A, Phenomenex, Germany) in a one-column setup. Following loading and desalting at a flow of 700 nl/min at a maximum of 220 bar of water in 0.1% acetic acid, separation of the peptides was achieved by the application of a binary non-linear 70 min gradient from 5 to 50% acetonitrile in 0.1% acetic acid at a flow rate of 300 nl/min. The LC was coupled online to an LTQ Velos Orbitrap mass spectrometer (Thermo Fisher, Germany) at a spray voltage of 2.4 kV. After a survey scan in the Orbitrap (R = 30,000), MS/MS data were recorded for the twenty most intensive precursor ions in the linear ion trap. Singly charged ions were not taken into account for MS/MS analysis. The lock mass option was enabled throughout all analyses.

Tryptic digests of the membrane samples were analyzed the same way as the cytosolic samples with the modification that the LC was coupled online to an LTQ Orbitrap Classic mass spectrometer (Thermo Fisher, Germany) and that only the five most intensive precursor ions were chosen for MS/MS analysis in the linear ion trap.

After mass spectrometric measurement, MS data were converted into the mzXML format by msconvert (*Chambers et al., 2012*) and subsequently subjected to database searching via Sorcerer (SageN, Milptas, CA) using SEQUEST (Thermo Fisher Scientific, San Jose, CA; version 27, revision 11) without charge state deconvolution and deisotoping performed (details below).

## Protein identification, validation, and quantitation

Two protein sequence databases were constructed to identify proteins. The first database—Reduced Database (RedDB)—contained a total of 52,546 protein sequences from *B. azoricus* and its symbionts. Of these, 7714 sequences were from the SOX symbiont, 5955 sequences from the MOX symbiont, and 38,877 from the host.

The second database—Incremented Database (IncDB)—contained a total of 242,947 protein sequences from *B. azoricus*, its symbionts, and from organisms phylogenetically related to the

symbionts and the host. 93,304 protein sequences were from relatives of the SOX symbiont, 26,622 from relatives of the MOX symbiont, and 123,021 from host-related bivalves. Redundant sequences were removed using the CD-hit-2D clustering algorithm (*Li and Godzik, 2006*) at 100% sequence clustering threshold. To determine false discovery rates (FDRs) of protein identification, both protein sequence databases were reversed and appended to the original databases as decoy sequences.

All MS/MS spectra from 1D-PAGE-LC-MS/MS experiments were searched against the two protein sequence databases using the SEQUEST (v.27, rev. 11) algorithm (*Eng et al., 1994*) with the following parameters: Parent Mass Tolerance, 0.0065 Da; Fragment Ion Tolerance, 1 Da; up to 2 missed cleavages allowed (internal lysine and arginine residues), fully tryptic peptides only, and oxidation of methionine as variable modification (+15.99 Da).

Scaffold (version 4.0.6.1, Proteome Software Inc., Portland, OR) was used for filtering and analysis of protein identifications. For the searches against the IncDB, protein identifications were filtered with the PeptideProphet and ProteinProphet algorithms implemented in Scaffold (*Ma et al., 2012*) using the following thresholds: 95% for peptides and 99% for proteins. For the searches against the RedDB, protein identifications were filtered at the peptide level using SEQUEST scores (XCorrs of at least 2.5 (charge +2) and 3.5 (charges >+2), DeltaCN >0.08) and by requiring that they contained at least two identified peptides.

Protein level FDRs for all database searches were determined according to the method of *Käll et al. (2008)* using the number of identified decoy sequences that passed the filtering thresholds. FDRs for searches against the IncDB were all <1.7% and for the RedDB <0.2%.

For relative quantitation of proteins, normalized spectral abundance factor values, which give the relative abundance of a protein in a sample in %, were calculated for each sample according to the method of *Florens et al. (2006)* (*Supplementary file 1I*).

