## [Decision Letter]

Thank you for submitting your work entitled “An arsenal of toxin-related genes in the genomes of beneficial symbionts from deep-sea hydrothermal vent mussels” for peer review at *eLife*. Your submission has been favorably evaluated by Ian Baldwin (Senior Editor), a Reviewing Editor, and three reviewers.

The reviewers have discussed the reviews with one another and the Reviewing Editor has drafted this decision to help you prepare a revised submission.

The manuscript by Sayavedra et al. reports three draft genomes (of which one was published in 2011 by Petersen et al.) from chemosynthetic symbionts of three deep-sea hydrothermal vent mussels. The symbiontic bacteria were found to have an unusually high number of potentially virulence-associated genes. In particular genes coding for toxins of the types YD and RTX/MARTX were identified. Proteome data provide support for this finding. The findings have interesting and potentially far-reaching implications for the evolution of symbioses, as well as it gives insights into possible mechanisms of host-microbe interactions.

The reviewers found that this is a generally well-executed paper on a little known symbiont group, the results from which will be of general interest to the symbiosis and bacterial genomics fields, and that a number of improvements and corrections should be made before acceptance of the manuscript. The issues identified by the reviewers will require a rather extensive revision. Please find the following synthesis that combines the issues identified from the three reviewers into a single review:

1) The interesting statement that “33% of their genes may have been acquired by HGT” (in the Abstract) is not sufficiently supported by data and discussion in the manuscript. According to Table 1, this number was only found in 1 of 3 draft genomes. How representative is this value?

2) In the third paragraph of the Introduction, the prominent role of the *Vibrio fischeri* symbiont toxin (TCT) and its role in initiating squid symbioses should be cited in this context.

3) In the subsection “Draft genome sequences of *Bathymodiolus* symbionts” and Table 1 were the 33% of foreign genes only found in BspSym? Does the symbol “-” mean that these were not tested or not found in BazSymB and BazSymA? If they were not found, one would then have to question the relevance of HGT for these systems. If they were not tested, one would need to ask why this seemingly important analysis was omitted. As a minor point, completeness estimates that are currently stated in the text should be included in this table.

4) In the subsection “Relationships to other toxins,” you state: “This cluster also contained a number of beneficial symbionts such as *Burkholderia rhizoxinica*, which is an intracellular symbiont of the fungus *Rhizopus*, and the *Photorhabdus* and *Xenorhabdus* symbionts of soil nematodes.” The examples should be correctly referenced. The *B. rhizoxinica* example is particularly interesting in this context (and very recently published in *eLife*, doi: 10.7554/I.03007) as this bacterial symbiont of a fungus known to cause rice blight could potentially share a similar strategy in symbiosis establishment.

5) In the subsection “Expression of toxin-related genes,” the section on transcriptome sequencing, as promised in the Abstract, is presented in the manuscript only with the first two sentences. There appears to be no visuals (figures, tables neither in main body nor supplement) and no methods description. The RNASeq approach should be removed.

6) In the subsection “Are some of the toxins used for direct beneficial interactions between hosts and symbionts?”, other examples from the symbiotic world should also be mentioned.

7) In the Conclusions (and also the Abstract), the interpretation of a tamed toxin, although a catchy phrase, appears overstated as long as the function of the toxins is not known. Is it not also conceivable that the symbionts are not as “beneficial” as previously thought? Provocatively stated, must the paradigm of *Bathymodiolus* symbioses be rewritten? The early window of host-symbiont contact or a system like the present one, where a symbiont must always be acquired horizontally anew with each new generation may need a specific/aggressive line of defense. Alternatively, and in analogy to the *Xenorhabdus* system, could the host be using the toxins for defense against bigger predators or could chemical defense play a role? Several scenarios are conceivable, which attribute to the relevance of the presented findings.

8) The Conclusions paragraph provides an elegant and thought-provoking perspective and a testimony why the findings are relevant. The results could indeed be paradigm-shifting in chemoautotrophic symbioses and other, related symbiosis-systems in general.

9) With regards the subsection “Sampling and processing of *Bathymodiolus* mussels”, where are the transcriptome data and why were they not included?

10) This is the first report of the genome sequence(s) of mussel chemosynthetic symbionts: “[…] to assemble the first essentially complete draft genomes of the SOX symbionts from *Bathymodiolus* mussels” (last paragraph of the Introduction). However, this paper “leaps” over characterization of the genomes to focus on a particular subset of genes with hardly any description of the genomes themselves. The only information included in addition to the “toxin” genes is the size, 16S identity, GC content, shared numbers of genes, and mobile elements. No details or discussion on the coverage (total bp vary widely for the 3 genomes, as only part of the 16S was recovered for one host symbiont, suggests not very good), binning (*B. azoricus* has two different symbionts – SOX, MOX), assembly (or lack thereof), sequence heterogeneity, meta genomes (if environmental – different strains?), and genome size (very small for a free-living symbiont if estimate is correct from contigs). Functional genes/metabolic capabilities are not described. Further, the reported aspects of the mussel genomes are only compared to the vesicomyids and SUP05. What about the other symbionts?

In the absence of a general description of the mussel symbiont genomes, it is hard to put the importance of “toxin” genes in context. If this was just another strain of a bacterial species, then perhaps it is okay to skip any real description of the genome. However, there are very few chemosynthetic symbiont genomes sequenced to date, only 7 in total: 2 tubeworms, 1 oligochaetes, and 4 molluscs: 2 vesicomyids, 1 solemyid, and 1 snail. For these mussel symbionts, is there any genomic evidence for free-living? Mobile elements? Heterotrophs? Actually, I was most surprised to find in Appendix 2 that “the symbionts appear to be obligate autotrophs […]” as all 3 genomes lack alpha-ketoglutarate DH, and have few transporters for organic C uptake.

Please provide the primary description of the genomes in the supplement including a cursory description of major metabolic pathways.

11) “Toxin” genes is the main focus of this paper, but the function of these genes is not known, and the genes were identified based only on their sequence annotation as “toxin” or “virulence” genes, falling into three classes based in part on shared motifs. The sequences are so divergent they cannot be confidently aligned except for a few of the YD class (in the subsection “Relationships to other toxins”). The authors go to great lengths and sophisticated analyses to ID them as “toxin” genes including constructing gene similarity networks with clusters containing genes with >25% similarity over at least half of the gene length. They compare the mussel symbiont toxin genes with those of other bacteria, examining “host-associated”, free-living”, etc. relationships with interesting correlations. Transcriptome analyses show they are all expressed, with some also via proteomics.

These analyses are quite interesting, and certainly challenge common thinking about mutualistic vs. parasitic associations and the roles of bacterial proteins and their role in disease. However, the functions of these “toxin” genes are not known, and the slant of the paper, including the title “toxin” and “arsenal” is almost entirely regarding the gene products as toxins.

“Toxin” genes are also detected in bacteria other than pathogens, also with unknown functions for the most part. The extensive discussion on the possible roles of these genes, necessitated given the lack of known function, is totally speculative and would benefit from a shorter and more focused treatment. For example, the authors speculate both that the bacteria have “tamed” the toxin genes to use in a mutualistic symbiosis and later that they may have been ancestral and their original function was mutualistic. Or that the toxin genes may function to protect the host against eukaryotic parasites.

On a semantic note, in efforts to promote studies of beneficial microbe/animal associations as informing parasitic or pathogenic associations, Margaret McFall-Ngai has suggested that terms like “virulence” and other disease related terminology be changed to more neutral terms, which I heartily endorse. The use of “toxin” or “pathogen” is disease-centric (and therefore mostly human-centric). To a bacterium, they have evolved mechanisms to obtain food (e.g., toxins to break open cells). If possible, use of a “symbiosis” term will be very helpful for presenting a balanced view of the possible functions of the “toxin” genes. Further, though “toxin-related” is used in the title, the paper quickly refers to these genes and their putative products as “toxins,” which, in the absence of function, is not correct.

12) Some larger patterns are suggested, e.g. that these genes are generally enriched in genomes of host associated bacteria. However, in the subsection “Are some of the toxins used for protection against eukaryotic parasites” and Figure 5, the analysis is not very meaningful because there is a phylogenetic component that is not taken into account. In other words, the clustering may reflect similarities of related species more than a repeated evolutionary pattern. This issue should be considered – can you show that there is a pattern independent of phylogeny?

[Editors' note: further revisions were requested prior to acceptance, as described below.]

Thank you for resubmitting your work entitled “Abundant toxin-related genes in the genomes of beneficial symbionts from deep-sea hydrothermal vent mussels” for further consideration at *eLife*. Your revised article has been favorably evaluated by Ian Baldwin (Senior Editor), a Reviewing Editor, and three reviewers. The manuscript has been considerably improved but there are some remaining minor issues that need to be addressed before acceptance, as outlined below:

The use of “toxins” vs. “toxin-related” is still a matter of debate. Some suggestions of Reviewer 2 should be considered. We also think that in general the use of toxin genes is over emphasized.

In the Abstract, the phrase “massive genome arrangement” was discussed which needs to be weakened in its statement.

An important question concerns the statement on RTX proteins forming pores in vacuolar members of any bacteria. Have any intracellular bacteria been shown to do such? If so, an example needs to be cited.

State in main text that the genomes are being described elsewhere.

Reviewer #1:

The authors have done a thorough job at addressing the reviewers’ concerns. In consequence, the manuscript is much improved, it is more streamlined and focused, and ambiguous wording has been removed.

I have no further comments at this point that would improve its quality and I therefore recommend publication.

Reviewer #2:

I am tremendously impressed with the authors' attention to the issues raised during review of the previously submitted manuscript, their changes to the manuscript, and their thoughtful responses. Virtually all of my earlier concerns have been addressed, but those which I feel still require attention are as follows:

1) I still find the term “toxin-related” to carry too heavy a connotation of the former. And as noted, we are trying to get “pathogen neutral” language used for host-microbe associations. What about using “toxin-related” but then abbreviate “T-R” and use that throughout. That way you have defined it, but the letters don't impress as negative.

Other toxins to fix:

a) “Array of toxins,” “these toxins” (Abstract).

b) “RTX toxins” (in the subsection “Gene-based comparison reveals toxin-related genes specific to *Bathymodiolus* symbionts”).

c) “[…] of toxin gene” (in the subsection heading “Statistical analyses of toxin gene content in microbial genomes”) – if it is shown in each case that it is a toxin, then the term is okay to use, otherwise, use “toxin and T-R genes[…]”.

d) “Toxin genes of the mussel” (in the subsection “Statistical analyses of toxin gene content in microbial genomes”).

2) Have the “toxin genes” been characterized as real toxins for organisms named? Including action, cyano, and firmicute? Clarify for any uncharacterized as “toxin-related.”

3) You refer to “massive genome rearrangements” (Abstract and text), but this is only relative to the related symbionts and SUP05. And in the subsection “General genome comparison,” couldn't these have been the ancestor and the others have undergone rearrangement? I suggest changing to more benign “extremely different” or “lack synteny” or other such descriptor.

4) Molecular mechanisms are extremely well described for Rhizobia. Clarify in sentence “animalhost microbe” (Introduction) or add in Rhizobia/legumes.

5) In the subsection “Are some of the toxin-related genes used for direct beneficial interactions between hosts and symbionts?” has it been shown that RTX proteins form pores in vacuolar members in any bacteria? Have any intracellular bacteria been shown to do such? If so, cite real examples. If not, this is a serious stretch and I favor removing. I searched the manuscript and there is very little on function of RTX.

6) I found the second paragraph of the subsection “Are some of the toxin-related genes used for protection against eukaryotic parasites?” quite interesting. As you have the MOX sequence(s), do they have the T-R genes?

7) State in main text that the genomes are being described elsewhere. Otherwise I still have the same reaction – why write about just toxin genes?

Reviewer #3:

The authors have done an excellent job in revising this manuscript. They have added several new analyses and some new data to address points raised by the reviewers.

It is a strong contribution, and I don't have further suggestions for improvement.

---

## [Author Response]

Essential revisions:

The reviewers found that this is a generally well-executed paper on a little known symbiont group, the results from which will be of general interest to the symbiosis and bacterial genomics fields, and that a number of improvements and corrections should be made before acceptance of the manuscript. The issues identified by the reviewers will require a rather extensive revision. Please find the following synthesis that combines the issues identified from the three reviewers into a single review.

*1) The interesting statement that “33% of their genes may have been acquired by HGT” (in the Abstract) is not sufficiently supported by data and discussion in the manuscript. According to*
Table 1*, this number was only found in 1 of 3 draft genomes. How representative is this value?*

Originally we only did this analysis for the genome that has the longest scaffolds, but you are correct, there is no reason why we shouldn’t do this analysis for all SOX symbiont draft genomes. We now added data from the missing two draft genomes of the SOX symbionts (see Table 1). The proportion of genes potentially acquired through HGT is very similar in all three, (30-35%).

*2) In the third paragraph of the Introduction, the prominent role of the* Vibrio fischeri *symbiont toxin (TCT) and its role in initiating squid symbioses should be cited in this context.*

Thank you for pointing out this example! We have now added the following paragraph to the Introduction:

“The symbiosis between *Vibrio fisheri* bacteria and *Euprymna scolopes* squid is one of the few beneficial host-microbe associations where the molecular mechanisms of host-symbiont interaction have been investigated. A number of factors are involved in initiating this symbiosis such as the symbiont-encoded ‘TCT toxin’, which is related to the tracheal cytotoxin of *Bordetella pertussis* (95).”

We also discovered another important example of toxin-like genes mediating symbiosis, and have added this information to the Discussion:

“MARTX-like genes also mediate cell-cell attachment in the symbiotic bacterial consortium ‘*Chlorochromatium aggregatum’* (88; 162).”

*3) In the subsection “Draft genome sequences of Bathymodiolus symbionts” and*
Table 1
*were the 33% of foreign genes only found in BspSym? Does the symbol “-” mean that these were not tested or not found in BazSymB and BazSymA?*

As above (response to point 1), we have now done this analysis for BazSymB and BazSymA and updated Table 1 accordingly.

If they were not found, one would then have to question the relevance of HGT for these systems. If they were not tested, one would need to ask why this seemingly important analysis was omitted. As a minor point, completeness estimates that are currently stated in the text should be included in this table.

We have now added new completeness estimates in Table 1. We used a new analysis tool, CheckM (Parks et al., PeerJ, 2015) to estimate the completeness of our draft genomes, which is between 90.7 to 97.7%.

*4) In the subsection “Relationships to other toxins,” you state: “This cluster also contained a number of beneficial symbionts such as* Burkholderia rhizoxinica*, which is an intracellular symbiont of the fungus* Rhizopus*, and the* Photorhabdus *and* Xenorhabdus *symbionts of soil nematodes.” The examples should be correctly referenced. The* B. rhizoxinica *example is particularly interesting in this context (and very recently published in eLife,* doi: 10.7554/I.03007*) as this bacterial symbiont of a fungus known to cause rice blight could potentially share a similar strategy in symbiosis establishment.*

Thank you for pointing out this excellent study. This section was shortened to remove redundancy in the manuscript. However, we kept a similar paragraph in the Introduction, which is now properly referenced as follows:

“Once they have been recognized, the symbionts need to enter host cells and avoid immediate digestion, just like other intracellular symbionts such as *Burkholderia rhizoxinica and Rhizobium leguminosarum,* or pathogens such as *Legionella, Listeria* or *Yersinia* (59; 100).”

5) In the subsection “Expression of toxin-related genes,” the section on transcriptome sequencing, as promised in the Abstract, is presented in the manuscript only with the first two sentences. There appears to be no visuals (figures, tables neither in main body nor supplement) and no methods description. The RNASeq approach should be removed.

We created a new table with the expression values of the toxins found in six metatranscriptomes (three animals each from the two different species) ([Supplementary-material SD2-data]). We have included a more detailed description of the transcriptomes in the Results section:

“Transcriptome sequencing revealed that all predicted toxin-related genes of the SOX symbionts in *B. azoricus* and *B*. sp gills were expressed. […] The expression levels of some genes from the RTX, MARTX and YD repeats classes were in some cases higher than the expression of the essential Calvin cycle gene ribulose bisphosphate carboxylase/oxidase (RuBisCO)*,* which accounted for 0.03 to 0.5% of mRNA in *B. azoricus* symbionts and 0.11 to 1.04% in *B*. sp. Symbionts.”

The transcriptomics methods are described in the subsection “Transcriptomics” of the Materials and methods section.

The RNAseq data is important not only for showing that the toxin-like genes are actually expressed by the symbionts in the host, but also for our analysis of the toxin gene variability within SOX symbiont populations. We would therefore prefer to keep this analysis in the manuscript.

6) In the subsection “Are some of the toxins used for direct beneficial interactions between hosts and symbionts?”, other examples from the symbiotic world should also be mentioned.

As written above in our response to point 4, this section has been shortened and removed from the Discussion, but we have now included some examples of beneficial symbioses to the similar paragraph in the Introduction.

7) In the Conclusions (and also the Abstract), the interpretation of a tamed toxin, although a catchy phrase, appears overstated as long as the function of the toxins is not known.

We have toned down the wording in the Abstract to simply say: “[…] we hypothesize that these toxin-related genes are used by the symbionts in beneficial interactions with their host.”

We have kept this wording in the Conclusions, as it leads into a discussion of the possibility that toxins were not ‘tamed’, but symbiosis factors evolved first and were subsequently ‘commandeered’ for use in pathogenesis. This paragraph was well regarded by the reviewers (see point 8).

*Is it not also conceivable that the symbionts are not as “beneficial” as previously thought? Provocatively stated, must the paradigm of* Bathymodiolus *symbioses be rewritten? The early window of host-symbiont contact or a system like the present one, where a symbiont must always be acquired horizontally anew with each new generation may need a specific/aggressive line of defense. Alternatively, and in analogy to the* Xenorhabdus *system, could the host be using the toxins for defense against bigger predators or could chemical defense play a role? Several scenarios are conceivable, which attribute to the relevance of the presented findings.*

We also considered the possibility that the symbionts are not as “beneficial” as previously thought, and had a short paragraph on this topic in Appendix 2 in the original submission. Considering the reviewers’ request to shorten the Discussion, we decided to keep this section in the Appendix. We have now improved the visibility of this Appendix section by beginning the Discussion as follows:

“There is overwhelming evidence that sulfur-oxidizing symbionts are beneficial for their *Bathymodiolus* mussel hosts (Appendix 2). It is therefore highly unlikely that the SOX symbionts are pathogens that have been mistaken for beneficial symbionts.”

As you correctly state, it is true that the symbiont could be defending the host against parasites (we hypothesize this role for some of the toxin-related proteins beginning in the Discussion section under the title “Are some of the toxins used for protection against eukaryotic parasites?”).

8) The Conclusions paragraph provides an elegant and thought-provoking perspective and a testimony why the findings are relevant. The results could indeed be paradigm-shifting in chemoautotrophic symbioses and other, related symbiosis-systems in general.

Thank you.

*9) With regards the subsection “Sampling and processing of* Bathymodiolus *mussels”, where are the transcriptome data and why were they not included?*

As mentioned above in our response to point 6, we now included a [Supplementary-material SD2-data] that includes the expression values. We have included more detail on the transcriptomics in the Results section (also see above in response to point 6).

*10) This is the first report of the genome sequence(s) of mussel chemosynthetic symbionts: “[…] to assemble the first essentially complete draft genomes of the SOX symbionts from* Bathymodiolus *mussels” (last paragraph of the introduction). However, this paper “leaps” over characterization of the genomes to focus on a particular subset of genes with hardly any description of the genomes themselves. The only info included in addition to the “toxin” genes is the size, 16S identity, GC content, shared numbers of genes, and mobile elements. No details or discussion on the coverage (total bp vary widely for the 3 genomes, as only part of the 16S was recovered for one host symbiont, suggests not very good), binning (*B. azoricus *has two different symbionts – SOX, MOX), assembly (or lack thereof), sequence heterogeneity, meta genomes (if environmental – different strains?), and genome size (very small for a free-living symbiont if estimate is correct from contigs). Functional genes/metabolic capabilities are not described. Further, the reported aspects of the mussel genomes are only compared to the vesicomyids and SUP05. What about the other symbionts?*

We added more information the genome coverage in Table 1. The details of the assembly and binning are described in the Materials and methods section under the heading “DNA extraction, sequencing, genome assembly and binning”. We included a sentence of what features Metawatt considers for the binning to make it clearer:

“454 sequencing was done by Genoscope to sequence the gradient pellet from gill tissue, and by OIST to sequence the adductor muscle of *B. azoricus*. […] The adductor muscle and gradient pellet metagenomes of *B. azoricus* were binned to separate the SOX symbiont from the MOX symbiont and host genomes with Metawatt V. 1.7, which uses tetranucleotide frequencies, coverage, GC content and taxonomic information for binning (156). Only sequences longer than 800 bp were considered for further analyses.”

Further, the reported aspects of the mussel genomes are only compared to the vesicomyids and SUP05. What about the other symbionts?

As above, our focus was on genome mining for genes that mediate host-microbe interactions in the *Bathymodiolus* symbiosis. We did this by comparing the genomes of the *Bathymodiolus* SOX symbionts with their closest free-living and symbiotic relatives that have a similar metabolism (autotrophic sulfur oxidizers), to identify genes specific to the *Bathymodiolus* SOX symbionts. A comparison of the genomic and metabolic features of all sulfur-oxidizing symbionts would be a fascinating project, but is beyond the scope of our study. To make this clearer, we added the following sentence to the final paragraph of our Introduction:

“The goal of this study was to identify the genomic basis of host-symbiont interactions in *Bathymodiolus* symbioses”.

*In the absence of a general description of the mussel symbiont genomes, it is hard to put the importance of “toxin” genes in context. If this was just another strain of a bacterial species, then perhaps it is okay to skip any real description of the genome. However, there are very few chemosynthetic symbiont genomes sequenced to date, only 7 in total: 2 tubeworms, 1 oligochaetes, and 4 molluscs: 2 vesicomyids, 1 solemyid, and 1 snail. For these mussel symbionts, is there any genomic evidence for free-living? Mobile elements? Heterotrophs? Actually, I was most surprised to find in*
Appendix 2
*that “the symbionts appear to be obligate autotrophs […]” as all 3 genomes lack alpha-ketoglutarate DH, and have few transporters for organic C uptake.*

Please provide the primary description of the genomes in the supplement including a cursory description of major metabolic pathways.

As you also correctly stated, our study focuses on one aspect of the symbiont genomes. Our collaborators from the University of Greifswald (Ruby Ponnudurai, Stephanie Markert, Thomas Schweder) are currently finishing a manuscript which will provide a much more detailed overview of the symbiont metabolism based on proteogenomic analyses. To provide more genomic context for the toxin-related genes in our manuscript, we included a new description of the key metabolic pathways encoded by the SOX symbiont genomes in Appendix 2. This also includes a new figure, which shows a metabolic reconstruction of the major pathways.

11) “Toxin” genes is the main focus of this paper, but the function of these genes is not known, and the genes were identified based only on their sequence annotation as “toxin” or “virulence” genes, falling into three classes based in part on shared motifs. The sequences are so divergent they cannot be confidently aligned except for a few of the YD class (in the subsection “Relationships to other toxins”). The authors go to great lengths and sophisticated analyses to ID them as “toxin” genes including constructing gene similarity networks with clusters containing genes with >25% similarity over at least half of the gene length. They compare the mussel symbiont toxin genes with those of other bacteria, examining “host-associated”, free-living”, etc. relationships with interesting correlations. Transcriptome analyses show they are all expressed, with some also via proteomics.

These analyses are quite interesting, and certainly challenge common thinking about mutualistic vs. parasitic associations and the roles of bacterial proteins and their role in disease. However, the functions of these “toxin” genes are not known, and the slant of the paper, including the title “toxin” and “arsenal” is almost entirely regarding the gene products as toxins.

Throughout the paper, we now use the correct and more conservative term ‘toxin-related’, which reflects their similarity to characterized toxins, but allows for the fact that their function in this symbiosis is so far unknown. We also made the title more neutral by changing it to “Abundant toxin-related genes in the genomes of beneficial symbionts from deep-sea hydrothermal vent mussels”.

“Toxin” genes are also detected in bacteria other than pathogens, also with unknown functions for the most part. The extensive discussion on the possible roles of these genes, necessitated given the lack of known function, is totally speculative and would benefit from a shorter and more focused treatment.

We have shortened the Discussion significantly. The revised version is more focused, and we have removed repetition and redundancy.

For example, the authors speculate both that the bacteria have “tamed” the toxin genes to use in a mutualistic symbiosis and later that they may have been ancestral and their original function was mutualistic. Or that the toxin genes may function to protect the host against eukaryotic parasites.

On a semantic note, in efforts to promote studies of beneficial microbe/animal associations as informing parasitic or pathogenic associations, Margaret McFall-Ngai has suggested that terms like “virulence” and other disease related terminology be changed to more neutral terms, which I heartily endorse. The use of “toxin” or “pathogen” is disease-centric (and therefore mostly human-centric). To a bacterium, they have evolved mechanisms to obtain food (e.g., toxins to break open cells). If possible, use of a “symbiosis” term will be very helpful for presenting a balanced view of the possible functions of the “toxin” genes.

We completely agree with the opinions of the reviewers and Margaret McFall-Ngai. In our manuscript, we chose to use the term ‘toxin-related’ to refer to these ‘symbiosis genes’. We think it is important that the term ‘toxin’ is used. Firstly, similar genes have proven functions as toxins. Secondly, the term ‘symbiosis genes’ may not yet be widely understood by researchers working on pathogenesis, and ideally we would like to get this message across to this field as well. The final paragraph of the Conclusions focuses on exactly this issue.

Further, though “toxin-related” is used in the title, the paper quickly refers to these genes and their putative products as “toxins”, which, in the absence of function, is not correct.

We agree and now call these genes “toxin-related” consistently throughout the manuscript.

*12) Some larger patterns are suggested, e.g. that these genes are generally enriched in genomes of host associated bacteria. However, in the subsection “Are some of the toxins used for protection against eukaryotic parasites” and*
Figure 5*, the analysis is not very meaningful because there is a phylogenetic component that is not taken into account. In other words, the clustering may reflect similarities of related species more than a repeated evolutionary pattern. This issue should be considered – can you show that there is a pattern independent of phylogeny?*

That is an excellent point. The first indication that phylogeny is not the main driver came from our observation that the toxins were distributed among very distantly related taxonomic groups. We have now done a more detailed analysis testing for enrichment of toxins depending on taxonomic classification at the order, class and family levels. YD repeats and MARTX were not enriched at any of these levels, but we did find that RTX genes were enriched at the order level.

Similar analyses showed a significant enrichment of YD genes in intracellular bacteria and MARTX in host-associated, but not of RTX. We therefore conclude that lifestyle rather than phylogeny is the main factor driving the distribution of these genes. We added the following text in the Results:

“Bacteria that are closely related often have similar genomic and physiological features. […] Therefore, phylogeny is not the main driver in the toxin distribution of YD repeats genes and MARTX.”

While doing this new analysis, we realized that there was some redundancy in the data used for our first Kruskal-Wallis and Mann-Whitney-Wilcoxon tests. We redid this analysis after removing some species that were represented twice in the dataset. This resulted in slightly different P values, however, the overall results stayed the same, and those categories that were significant in our first analysis are still significant.

[Editors' note: further revisions were requested prior to acceptance, as described below.]

[…] Reviewer #2:

I am tremendously impressed with the authors' attention to the issues raised during review of the previously submitted manuscript, their changes to the manuscript, and their thoughtful responses. Virtually all of my earlier concerns have been addressed, but those which I feel still require attention are as follows:

1) I still find the term “toxin-related” to carry too heavy a connotation of the former. And as noted, we are trying to get “pathogen neutral” language used for host-microbe associations. What about using “toxin-related” but then abbreviate “T-R” and use that throughout. That way you have defined it, but the letters don't impress as negative.

Other toxins to fix:

a) “Array of toxins,” “these toxins” (Abstract).

*b) “RTX toxins” (in the subsection “Gene-based comparison reveals toxin-related genes specific to* Bathymodiolus *symbionts”).*

c) “[…] of toxin gene” (in the subsection heading “Statistical analyses of toxin gene content in microbial genomes”) – if it is shown in each case that it is a toxin, then the term is okay to use, otherwise, use “toxin and T-R genes[…]”.

d) “Toxin genes of the mussel” (in the subsection “Statistical analyses of toxin gene content in microbial genomes”).

Thanks for your suggestions. We are on the same page about using neutral terms. The term ‘Symbiosis genes’ would be preferable, but without experimental evidence to support the role of these genes in the symbiosis, we prefer the more cautious term ‘toxin-related genes’. We now use the abbreviation TRG for toxin-related gene throughout the manuscript. We also changed all of the above listed to TRGs. Thank you for pointing out the few occasions where the term ‘toxin gene’ slipped through after our first round of revisions.

2) Have the “toxin genes” been characterized as real toxins for organisms named? Including action, cyano, and firmicute? Clarify for any uncharacterized as “toxin-related.”

We modified the text as follows:

“TRGs from the *Bathymodiolus* symbionts clustered together with toxins and TRGs from phylogenetically diverse organisms including characterized toxins of gammaproteobacterial *Vibrio* and *Pseudomonas*, and TRGs of the gammaproteobacteria *Shewanella*, the actinobacterial *Rhodococcus,* the cyanobacterium *Trichodesmium*, and the firmicute *Caldicellulosiruptor.”*

3) You refer to “massive genome rearrangements” (Abstract and text), but this is only relative to the related symbionts and SUP05. And in the subsection “General genome comparison,” couldn't these have been the ancestor and the others have undergone rearrangement? I suggest changing to more benign “extremely different” or “lack synteny” or other such descriptor.

We think that the SOX symbiont genomes is under constant genome reshuffling based on the following evidence: 1) The SOX *Bathymodiolus* symbiont has a higher abundance of mobile elements compared to their close relatives, 2) Mobile elements are among the top 50 most expressed genes 3) Comparison of the two SOX symbionts of *B. azoricus* show genome rearrangements even between these two closely-related symbionts. These observations suggest that genome reshuffling is a significant and undergoing process in the SOX symbiont genomes. Nevertheless, we toned down the terms in the text as follows:

“Comparison of these symbiont genomes with those of their closest relatives revealed that the symbionts have undergone genome rearrangements, and up to 35% of their genes may have been acquired by horizontal gene transfer.” (Abstract)

“BspSym, the genome that assembled into the fewest contigs, lacked synteny compared to SUP05 and the clam symbionts, as shown by whole-genome alignment (Figure 2—figure supplement 2). […] These regions without genome synteny therefore most likely represent true genome reshuffling in *Bathymodiolus* symbionts.” (Subsection “General genome comparison”)

“The lack of synteny we observed in the symbiont genomes is consistent with the presence of mobile elements and major HGT events (1; 74; 133).” (Subsection “Origin of TRGs in Bathymodiolus symbionts”)

Subsection heading: “PCR amplification of regions with lack of synteny”.

“Figure 2—figure supplement 2. Whole genome alignment. Each colored block is a region of the genome that aligned to part of another genome because it is homologous and the genes are arranged in the same order.” (Figure 2—figure supplement 2 legend)

4) Molecular mechanisms are extremely well described for Rhizobia. Clarify in sentence “animalhost microbe” (Introduction) or add in Rhizobia/legumes.

We modified the text according to your suggestion:

“Although the mechanisms of host cell entry and immune evasion have been extensively studied in pathogens and plant-microbe associations such as the Rhizobia-legume symbiosis, far less is known about the mechanisms beneficial symbionts use to enter and survive within animal host cells.”

5) In the subsection “Are some of the toxin-related genes used for direct beneficial interactions between hosts and symbionts?” has it been shown that RTX proteins form pores in vacuolar members in any bacteria? Have any intracellular bacteria been shown to do such? If so, cite real examples. If not, this is a serious stretch and I favor removing. I searched the manuscript and there is very little on function of RTX.

We removed this section.

6) I found the second paragraph of the subsection “Are some of the toxin-related genes used for protection against eukaryotic parasites?” quite interesting. As you have the MOX sequence(s), do they have the T-R genes?

This is indeed an interesting question. The MOX symbiont genomes do not have the toxin-related genes of the SOX symbiont. We added the following text:

“If many of the TRGs encoded by the *Bathymodiolus* SOX symbiont are being used to defend its host against parasites, as is hypothesized for *Ca*. H. defensa, then this could help to explain why *B. childressi* is so heavily infected by trematodes. The methane-oxidizing symbionts of *B. azoricus* and *B. childressi* do not encode TRGs (Antony CP, personal communication, May 2015).”

7) State in main text that the genomes are being described elsewhere. Otherwise I still have the same reaction – why write about just toxin genes??

We added the following text in the Results section:

“The core metabolic potential of the *Bathymodiolus* SOX symbionts is described in Appendix 1 – Symbiont Metabolism. A detailed description of the genomes is beyond the scope of this paper and will be published elsewhere”.